# Knowledge Accumulation in Continually Learned Representations and the Issue of Feature Forgetting

**Timm Hess**[*]  *timmfelix.hess@kuleuven.be*
*KU Leuven, Belgium*

**Eli Verwimp**[*]  *eli.verwimp@kuleuven.be*
*KU Leuven, Belgium*

**Gido M. van de Ven**  *gido.vandeven@kuleuven.be*
*KU Leuven, Belgium*

**Tinne Tuytelaars**  *tinne.tuytelaars@kuleuven.be*
*KU Leuven, Belgium*

**Reviewed on OpenReview:** *https://openreview.net/forum?id=aHtZuZfHcf*

## Abstract

Continual learning research has shown that neural networks suffer from catastrophic forgetting "at the output level", but it is debated whether this is also the case at the level of learned representations. Multiple recent studies ascribe representations a certain level of innate robustness against forgetting – that they only forget minimally in comparison with forgetting at the output level. We revisit and expand upon the experiments that revealed this difference in forgetting and illustrate the coexistence of two phenomena that affect the quality of continually learned representations: knowledge accumulation and feature forgetting. Taking both aspects into account, we show that, even though forgetting in the representation (*i.e.* feature forgetting) can be small in absolute terms, when measuring relative to how much was learned during a task, forgetting in the representation tends to be just as catastrophic as forgetting at the output level. Next we show that this feature forgetting is problematic as it substantially slows down the incremental learning of good general representations (*i.e.* knowledge accumulation). Finally, we study how feature forgetting and knowledge accumulation are affected by different types of continual learning methods.

## 1 Introduction

Machine learning models are typically trained on static datasets, and once they are deployed, they are usually not updated anymore. However, sometimes models make mistakes. Sometimes they do not work in a domain that was not trained on. Sometimes they do not recognize certain classes or corner cases. To overcome such malfunctions, the default choice in industry is to gather new data and to retrain a model from the beginning with both the new and old data (Huyen, 2022). Retraining a full model is costly and time-consuming, especially in deep learning. The goal of continual learning is to enable models to train continually, to learn from new data when they become available. This has proven to be a hard challenge (De Lange et al., 2022; van de Ven et al., 2022), as deep learning models that are continually trained exhibit catastrophic forgetting (McCloskey & Cohen, 1989; Ratcliff, 1990). Without precautionary measures, new information is learned at the expense of forgetting earlier acquired knowledge.

The data to train machine learning models rarely come in a format that is adapted to the problem one intends to solve. Taking the example of visual data, it is near impossible to infer high-level properties directly from

---

[*]Authors contributed equally.

an image's raw pixel values. Hence, a first step is usually to transform the data into a *representation* that makes solving the problem at hand easier. Often deep neural networks are used for this (Bengio et al., 2013). These networks learn semantically meaningful representations while optimizing their parameters to learn an input-output mapping. Sometimes the goal is the representation itself, yet often it is a final layer, or head, that uses the learned representation to assign an output (*e.g.* a class label) to an input. Often the backbone network used to calculate a representation and the head that uses this representation are trained in unison, but it can be useful to think of them as two separate entities working together.

In continual learning, there are at least two good reasons to care about representations. First, a strong representation makes it easier to learn new information (Kornblith et al., 2019). When a model already has a good representation, it may require less changes to adapt to new data, which lowers the risk of forgetting (Cha et al., 2021). Second, continually learning a good representation by progressively accumulating knowledge from individual tasks is a goal on its own. Effective continual learning should be able to use new information to its benefit and build a stronger representation over time (Chen & Liu, 2018), which can finally be used to solve a variety of tasks (Bengio et al., 2013).

It is with these motivations that recent works have been studying how representations are learned in continual learning, and how they forget. Multiple studies observe an apparent robustness to forgetting for representations (Davari et al., 2022; Zhang et al., 2022; Hu et al., 2022; Wang et al., 2023). For example, Davari et al. (2022) write: "*[...] in many commonly studied cases of catastrophic forgetting, the representations under naive fine-tuning approaches, undergo minimal forgetting, without losing critical task information*" (p. 16713), and Zhang et al. (2022) comment: "*there seems to be no catastrophic forgetting in terms of representations*" (p. 4) and "*common techniques for mitigating catastrophic forgetting [...] have little effect on improving [representations]*" (p. 4). These studies further suggest that in the case that some forgetting of learned features does happen, this does not hinder the learning of good general representations. For example, when measuring the representation quality for a task that was not included in the training sequence, Zhang et al. (2022) conclude that "*representation learning and catastrophic forgetting are largely separate issues*" (p. 9). If this is true, forgetting would only be a problem if one cares about the performance on the trained tasks, but not if one cares about learning a good general representation.

In this work, we carefully revisit the dynamics of learning and forgetting at the representation level, and we illustrate where its apparent innate robustness to forgetting originates. First, we highlight two key phenomena affecting the quality of the continually learned representation: feature forgetting and knowledge accumulation. Then, by considering both jointly, we re-investigate continual learning at the representation level. In short, we try to answer two questions:

**Question 1**: *Do continually trained representations forget catastrophically?*

With extensive experiments we show that, also at the level of representations, newly learned information tends to be rapidly and drastically forgotten during continued training on other tasks. The first tasks seemingly forget less, yet we show that the non-forgotten information is information that is shared between tasks. This leads us to the follow-up question:

**Question 2**: *What are the consequences of forgetting these representations?*

To determine the impact of feature forgetting on the quality of the continually learned representation for downstream tasks, we compare the representation of a continually trained model against a representation that is ensembled from copies of the model obtained after finishing training on each task. This ensemble baseline and the continually trained model learn in the exact same way, but differ because the former does not forget. We find that the ensemble baseline yields a substantially better general representation than the continually trained model, which indicates that feature forgetting slows down knowledge accumulation (*i.e.* the improvement of the learned representation over time). This means that preventing feature forgetting is not only important for the performance on tasks that a model was trained on, but also to learn strong representations in general.

Most experiments in this paper study the continual learning and forgetting of representations in supervised learning, though in Appendix C we show evidence that suggests that our main conclusions also hold for

self-supervised learning. We conclude the paper by evaluating examples of important families of continual learning methods and reporting how they influence the learning and forgetting of representations.

In summary, our contributions include[1] :

- We illustrate the characteristics of feature forgetting and knowledge accumulation evaluated at the representation level (Section 3)

- We show that newly learned information is forgotten just as catastrophically at the representation level as it is at the output level, except when the information can also be learned from other tasks (Sections 4.1 and 4.2).

- We show that forgetting in the representation hinders knowledge accumulation (Section 4.3).

- We compare the level of feature forgetting and knowledge accumulation for different types of continual learning methods (Section 5).

## 2 Preliminaries

We follow the common definition of a continual learning setting by assuming a sequence of $m$ disjoint tasks $\mathcal{T} = \{T_1, T_2, ..., T_m\}$. Each task $T_j$ consists of training data with inputs $X_j$ and targets $Y_j$, as well as respective test data $\hat{X}_j$, $\hat{Y}_j$. While training on task $T_j$, only the training data $\{X_j, Y_j\}$ of that task are used. Exceptions are replay memories, which can store small subsets of data from past tasks; we only use such replay memories when studying experience replay in Section 5. Finally, we use $T_d$ to refer to a *downstream task, i.e.* one that is not part of the training sequence $\mathcal{T}$. To study the representation learned by a model on the sequence $\mathcal{T}$, we distinguish between the backbone parameters $\theta_B$ and the task-specific heads $\theta_H = \{\theta_{h_1}, ..., \theta_{h_m}\}$. For task $T_j$, the output of a model is calculated as $g(f(x; \theta_B); \theta_{h_j})$ and its representation as $f(x; \theta_B)$. In this paper, each head $\theta_{h_j}$ specifies a linear layer and we use, with slight abuse of notation, $f(x; \theta_{h_j}, \theta_B)$ to refer to the combination of backbone and head. Alternatively, we use $f_i$ to denote the entire model directly after training on task $T_i$, and we use $f_0$ for a model with random weights.

Our main focus is on classification tasks. We use $r_{i,j}^f$ as a general symbol to refer to a metric that quantifies the performance or quality of model $f_i$ for task $T_j$. We drop super- and subscript when they are clear from context. To differentiate between the quality of the backbone and the entire model, we define two such performance metrics: *output accuracy* $\text{OUT}_{i,j}$ and *linear probe accuracy* $\text{LP}_{i,j}$ (or LP accuracy). $\text{OUT}_{i,j}$ refers to the standard test accuracy of model $f(x; \theta_{h_j}, \theta_B)$ immediately after training task $T_i$. When $i < j$, output accuracy is equal to random performance, *i.e.* the performance of randomly initialized head. To measure the quality of a representation, one option would be to use the optimal head $\theta_{h_j}^*$ for a particular backbone $\theta_B$ and task $T_j$. To approximate this optimal head, we train a new head with parameters $\tilde{\theta}_{h_j}$ using the training data of task $T_j$ while keeping the backbone parameters $\theta_B$ frozen, as is common in representation learning (Bengio et al., 2013). Using this new head, $\text{LP}_{i,j}$ is equal to the test accuracy of $f(x; \tilde{\theta}_{h_j}, \theta_B)$ on task $T_j$, whereby $\theta_B$ are the backbone parameters immediately after training on task $T_i$. Alternative ways to evaluate representations reach similar results for the experiments reported in this paper, see Appendix H.

In the main paper, the reported results are on Split MiniImageNet, a 20 task (five classes each) split of MiniImageNet (Vinyals et al., 2016). We use 19 tasks as the training sequence $\mathcal{T}$, while the remaining task is never seen during training and used as downstream task $T_d$. To reduce the influence of the difficulty of a task, we use five task splits and report mean and standard error (SE) on all results. The splits are randomly generated but consistent across experiments. When evaluating the performance on a task, the model only needs to choose between the classes of that task. We do not include all classes seen so far (i.e., as would be done in a class-incremental learning evaluation), because this would make the evaluation 'task' more difficult after each trained task, thereby confounding the analysis. Further details of our experimental protocol are in Appendix A. In Appendix G, we report experiments using Split CIFAR-100 (Krizhevsky et al., 2009).

---

[1]Code available at: `https://github.com/TimmHess/KAaFF`

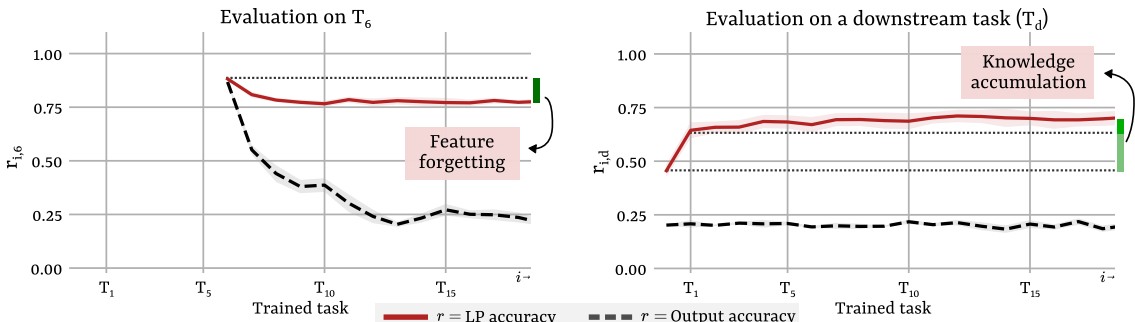

Figure 1: Illustration of feature forgetting and knowledge accumulation, for a model continually finetuned on Split MiniImageNet. On the left, the output accuracy and LP accuracy on $T_6$ are shown. The difference between LP accuracy directly after the task was trained and the LP accuracy after training on later tasks is what we call feature forgetting. On the right, the same continually trained model is evaluated with the same metrics but on a downstream task ($T_d$). We refer to the improvement in LP accuracy over that of a randomly initialized model ($f_0$) as knowledge accumulation. (Mean $\pm$ SE, 5 runs)

## 3 Knowledge accumulation and feature forgetting

Catastrophic forgetting is observed when measuring accuracy at the output level of a continually finetuned neural network. However, recent studies exemplify how evaluating performance of the same model at representation level instead leads to a different result. In the following, we highlight two evaluation perspectives that have been used in recent literature to illustrate this difference between the output and representation level. With these evaluation perspectives, two phenomena can be identified that affect the quality of continually learned representations: *feature forgetting* and *knowledge accumulation*. The evaluation perspectives differ in what data they evaluate on. On the one hand, a task that was included in the training sequence (*e.g.* $T_6$) can be used, on the other hand, a downstream task ($T_d$) that is excluded from the training sequence can be used. Each perspective can show one of the two phenomena.

Evaluating on a task included in the training sequence (*e.g.* $T_6$), it can be observed that linear probing, *i.e.* retraining the head, can recover much of the performance that is lost in the output accuracy (Figure 1, left), which could be interpreted as representations forgetting less than at the output (Davari et al., 2022). The performance degradation, or forgetting, that persists beyond the recovered performance of the linear probe is what we call *feature forgetting*. When features are forgotten, the representations of past classes become less separable. This contributes to the total amount of forgetting in continual learning, to which *e.g.* representation drift and misaligned heads contribute as well (Caccia et al., 2021), see also Section 7.

The second evaluation perspective inspects the performance of a continually trained model on a downstream task. Using the same setting as above, we find that the representation for $T_d$ progressively improves with each learned task, without significant declines (Figure 1, right). We call this improvement of a representation for unseen tasks *knowledge accumulation*. A model that has accumulated more knowledge by this definition, is a more general model and is better suited to solve new tasks, although it may have forgotten features specific to a past task at the same time. This definition is similar to the one used by Jin et al. (2021), who equate knowledge accumulation with the capability of generalizing to unseen tasks. Lesort et al. (2023) and Caccia et al. (2020) use the term knowledge accumulation slightly differently, to loosely refer to performance on both old and new tasks. Knowledge accumulation is related to forward transfer, which is one of the main desiderata in task-incremental learning (van de Ven et al., 2022; De Lange et al., 2022). Yet task-incremental learning results themselves are not well suited to study the representations directly, as representation drift (Caccia et al., 2021) can result in misaligned heads, which could suggest that there is no knowledge accumulation, even when there is. Finally, in continual pretraining (Hu et al., 2022; Lee et al., 2023; Cossu et al., 2022), the performance on downstream tasks is what matters, yet the influence of forgetting pretrained features has not been studied in these works. In what follows, we use both concepts, feature forgetting and knowledge accumulation, to study the dynamics of continual learning at the representation level.

## 4 Forgetting in continually learned representations

In this section, we analyze feature forgetting by taking knowledge accumulation into account. Section 4.1 shows that newly gained performance tends to be lost similarly at the output and the representation. Features learned during the first tasks in the training sequence are an exception to this, which we investigate further in Section 4.2. Finally, Section 4.3 points out that feature forgetting hinders effective knowledge accumulation for downstream tasks.

### 4.1 Relative forgetting

McCloskey & Cohen (1989) and Ratcliff (1990) are often credited for discovering the phenomenon of catastrophic forgetting. They describe it respectively as: "*[t]raining on a new set of items may drastically disrupt performance on previously learned items*" (p. 110), and "*well-learned information is forgotten rapidly as new information is learned*" (p. 285). In recent literature, following Lopez-Paz & Ranzato (2017), forgetting is commonly quantified as the difference between the performance immediately after task $T_j$ was trained on and the performance after training on $n$ new tasks: $r_{j,j} - r_{j+n,j}$. Both output accuracy and linear probing accuracy have been used in this way (*e.g.* Davari et al., 2022). Forgetting, defined as such, is thus equal to the absolute drop in performance. In contrast, the amount of forgetting can also be considered relative to how much information was learned *during* training on the task. We propose to define the *relative forgetting* of task $T_j$ after $n$ new tasks as:

$$\mathrm{rF}_{n,j}^r = \frac{r_{j,j} - r_{j+n,j}}{r_{j,j} - r_{j-1,j}} \tag{1}$$

Or, in words: relative forgetting measures how much of the performance that was gained during training on a task is forgotten afterwards. Figure 2 illustrates how absolute forgetting and relative forgetting can lead to different conclusions. When inspecting the absolute forgetting at the output and the representation (middle panel), one could conclude that the forgetting at the output is worse than at the representation level. However, the representation quality also improved less during training of task $T_6$ (left panel), partly because of previously accumulated knowledge. Relative to the gained performance, forgetting is roughly equal at the representation and at the output (right panel); both lose nearly all gained performance. Considering the gained performance is thus crucial when comparing forgetting in settings where the performance *before* training a task differs across the compared settings, because it changes the denominator of Equation 1. This happens when considering representations and LP accuracy, but in most cases not for output accuracy. Their difference is comparable to the difference between the answers to the following two questions before seeing any task data: *Which test samples belong to the unknown category x?* and *Given that x looks like this, which other test samples are of category x?*. While the first answer will be random, the second one depends on how good the description, *e.g.* the representation, of $x$ is. A model that has accumulated more knowledge and has a better representation can have improved performance on the second question, but not on the first.

Figure 3 shows the relative forgetting of the representation and of the output for tasks 1 until 9 (the one of $T_6$ is thus a copy of the right panel in Figure 2). For early tasks (especially $T_1$), forgetting as measured by linear probing stabilizes after one or two new tasks, while for the output $\mathrm{rF}_{n,j}$ continues to increase. For the later tasks, the representation forgets at least as much as at the output (see Appendix F for task $T_{10}$ and more). This observation noticeably leads to a different conclusion than the one drawn by the papers in the introduction: except for the first tasks, almost all performance that was gained while training a task, is lost when training on new ones, both at the output and the representation level.

### 4.2 Exclusion baseline

The previous section showed that nearly all performance gained during a task tends to be quickly lost both for the representation and at the output, except for the first tasks in the training sequence. For these tasks, there is a significant difference between the relative forgetting at the output and the representation, see Figure 3. After training a few new tasks, the output has $\mathrm{rF} \approx 1$, while the representation has a significantly lower rF. In this subsection, we show that this difference can be explained by knowledge accumulation from

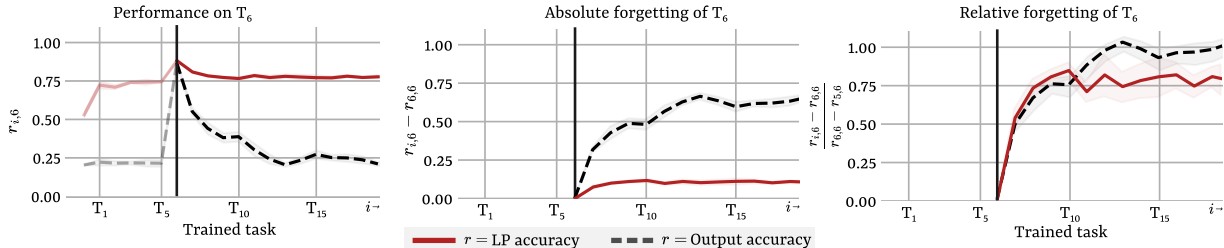

Figure 2: Performance, absolute forgetting and relative forgetting of $T_6$ for a model continually finetuned on Split MiniImageNet. Comparing output accuracy and representation quality (LP accuracy) can have different interpretations when considered in absolute or relative terms. Absolute forgetting (middle) suggests that output forgetting is worse than at the representation level. Such absolute forgettting does not account for performance already accumulated before learning $T_6$ (left). When expressing forgetting *relative* to newly gained performance (right), forgetting trends are similar at the output and representation level. (Mean ± SE, 5 runs)

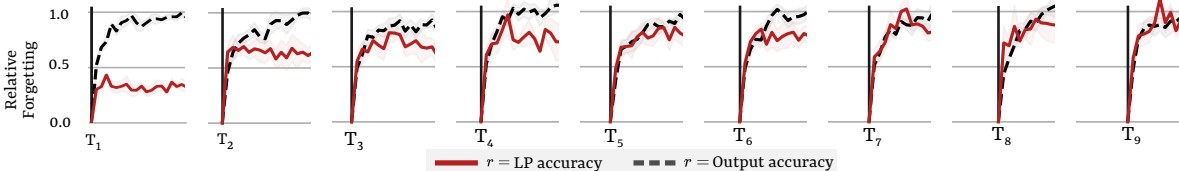

Figure 3: Relative forgetting at the output level and at the level of representations, as per Eq. 1, for a model continually fine-tuned on the Split MiniImageNet sequence. Relative to the knowledge gained during training on the task, forgetting in the representation and at the output are similar, except for the first few tasks. (Mean ± SE, 5 runs)

other tasks. When learning one task improves the representation for another, those tasks have something in common. Even if a model forgets everything it had learned, the common information can be re-learned from the later tasks. After training on all tasks, the final model would still perform better than a model with random weights, which results in rF scores lower than one. To further unravel how knowledge accumulation influences forgetting, we train a new model $f^{-j}$ on the same task sequence as in the previous section, but *without* task $T_j$, see Figure 4a. Model $f^{-j}$ cannot forget specific information on task $T_j$ as it was never trained on it, but it can accumulate common knowledge from other tasks. We define $\text{EXC}_{i,j}$ as the difference between both in models in LP accuracy for task $T_j$ after training task $T_i$:

$$\text{EXC}_{i,j} = \text{LP}_{i,j}^{f} - \text{LP}_{i,j}^{f^{-j}} \tag{2}$$

If this difference is larger than zero, model $f$ remembers some specific information that was not learned from other tasks. Figure 4 and Table 1 show that for all tasks, the difference reduces to almost zero. The information that was not forgotten from the earlier tasks, could thus also be learned from other tasks and is not specific to that task only. Table 1 suggests that there might be a subtle difference at the end of the training sequence between $f$ and $f^{-j}$, but the errors are too large to draw definite conclusions.

Learning one task does, in most cases, not improve the output accuracy of another. See *e.g.* the left panel of Figure 2, while the LP accuracy for $T_6$ improves even before $T_6$ is trained, the output accuracy remains at chance level. Forgotten information thus cannot be re-learned from other tasks, which suggests why the relative forgetting at the output does approach one. Similar results are to be expected if the learned representation for one task does not improve the other. This observation implies an important nuance: even when the output accuracies of two models are both at chance level, one can have a better representation than the other for a particular task.

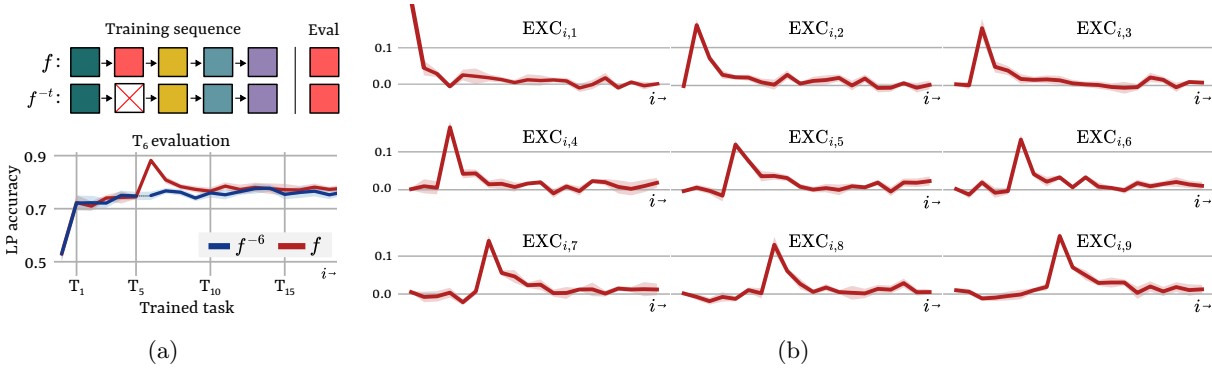

Figure 4: Illustration of the task exclusion difference (EXC, Eq. 2) for continual finetuning on Split MiniImageNet. Panel (a) depicts the adjusted training sequence for the "exclusion model" $f^{-j}$, and a comparison of the resulting LP accuracy for continually finetuned model $f$ and $f^{-6}$ on task $T_6$. Panel (b) shows that $\text{EXC}_{i,j}$, for the first 9 tasks, always follows the same trend. It averages around zero for tasks $i < j$, peaks for $i = j$, and quickly reduces to almost zero again for $i > j$ (Mean $\pm$ SE, 5 runs).

Table 1: Quantitative results of Figure 4b. The final difference ($\text{EXC}_{19,j}$) and maximal difference ($\text{EXC}_{j,j}$) between model $f$ and its exclusion counterpart $f^{-j}$. (Mean $\pm$ SE, 5 runs)

| $T_j$ | $T_1$ | $T_2$ | $T_3$ | $T_4$ | $T_5$ | $T_6$ | $T_7$ | $T_8$ | $T_9$ |
|---|---|---|---|---|---|---|---|---|---|
| $\text{EXC}_{19,j}$ | $0.001 \pm 0.004$ | $0.002 \pm 0.011$ | $0.008 \pm 0.004$ | $0.018 \pm 0.013$ | $0.023 \pm 0.012$ | $0.010 \pm 0.010$ | $0.012 \pm 0.015$ | $0.005 \pm 0.007$ | $0.012 \pm 0.012$ |
| $\text{EXC}_{j,j}$ | $0.234 \pm 0.024$ | $0.159 \pm 0.015$ | $0.151 \pm 0.025$ | $0.165 \pm 0.015$ | $0.119 \pm 0.010$ | $0.132 \pm 0.012$ | $0.140 \pm 0.014$ | $0.130 \pm 0.021$ | $0.153 \pm 0.008$ |

### 4.3 Feature forgetting reduces knowledge accumulation

Previous sections showed that feature forgetting severely impacts newly learned features in neural networks and that what is not forgotten can also be learned from other tasks. This leads us to the second question of the introduction: what are the consequences of feature forgetting? Is the forgotten information important for that specific task only, or can it also contribute to more general knowledge accumulation? Phrased differently, how much knowledge accumulation would there be if there was no forgetting? To answer this, we consider a baseline that learns continually but has no forgetting, inspired by Vogelstein et al. (2020) and Yan et al. (2021). This *ensemble* baseline stores a model copy after every task. During evaluation, their representations are concatenated and a linear probe is trained on top of this concatenation. This ensemble learns exactly the same way as a continually finetuned model, but it cannot forget, as the task's original representation remains intact and can always be recovered by a head. See Figure 5a for a schematic of the approach, and Figure 5b for the results. Finetuning and the ensemble are exactly the same after training the first task. Knowledge accumulation from the first task aside, our ensemble baseline accumulates almost three times as much knowledge, signified by a 27.4% improvement on a downstream task versus only 9.3% for finetuning. It is interesting that finetuning also accumulates some knowledge during later tasks, albeit far inferior to what is possible when forgetting nothing.

Models trained on all tasks concurrently (jointly) rather than sequentially can also lead to improved representations (*e.g.* Zhang et al., 2022; Cha et al., 2022). This does not unambiguously lead to the same conclusion as our experiment. Besides having no forgetting, concurrent training can improve the representation for each task itself since it can contrast a single class with a wider variety of other classes. With our ensemble strategy, training remains task-per-task and the difference in performance with finetuning is only a consequence of forgetting nothing. Evaluation of the ensemble requires linearly more compute with every task, so we use it as handy tool to examine what would happen if a method does not forget but do not propose it as a new method. The larger dimension of the ensemble representation can be potentially confounding. In Appendix B we control for this using PCA dimension reduction and reach the same conclusion.

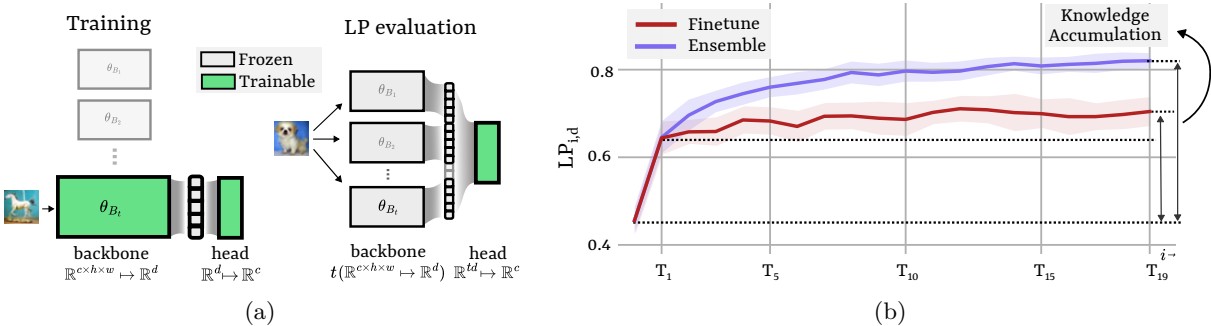

Figure 5: Schematic and result of the ensemble baseline. Panel (a) shows that training the ensemble baseline is the same as continual finetuning, but after every training phase a model copy is stored. During evaluation each stored copies produces a separate representation. Their representations are then concatenated and a linear classifier is trained on top of the concatenation. Panel (b) compares the LP performance of ensemble and finetuning on a downstream task. (Mean $\pm$ SE, 5 runs)

One task improving the performance of another, an effect of knowledge accumulation, is often named as a desideratum of continual learning (Chen & Liu, 2018), yet it persists as a difficult problem. Better representations should also lead to better continual learners. If one representation is better than another for data of a new task, it likely needs to change less to reach the same performance after training, thereby reducing the risk of forgetting. With few samples (*e.g.* a replay memory), a strong representation can also be used to quickly recover past information, sometimes referred to as 'fast remembering' (Hadsell et al., 2020; Davari et al., 2022). However, in Appendix D we re-emphasize that even a strong pretrained representation does not always imply mitigation of forgetting. An important step towards accomplishing these goals is preventing feature forgetting as much as possible without preventing models to learn.

The last sections illustrated and explained how addressing (feature) forgetting without considering knowledge accumulation and vice versa, can make representations appear to be robust against forgetting. Combining both offers a more nuanced story. Representations forget newly learned information as much at the representation level as at the output level, except for information that is shared across tasks. Preventing feature forgetting, as happens in our ensemble, presents an opportunity to increase knowledge accumulation. The next section will look at how some established continual learning methods seize this opportunity.

## 5    Can feature forgetting be prevented?

Over the last years, many methods to alleviate forgetting have been proposed. In this section, we review examples of some of the main families of methods and evaluate how they deal with feature forgetting and knowledge accumulation. The choice of algorithms is not driven by finding the best possible method, but we try to cover the most central ideas, in their simplest form. We test replay with a simple experience replay algorithm with 20 samples per class (ER), parameter regularization using Memory Aware Synapses (MAS) (Aljundi et al., 2018), functional regularization with Learning without Forgetting (LwF) (Li & Hoiem, 2017) and PackNet (Mallya & Lazebnik, 2018) as an architectural method. More variations on the standard replay algorithm are reported in the Appendix.

Figure 6 shows the relative forgetting of the tested methods and Figure 7 the results of the exclusion baseline for task $T_6$. Additionally, Table 2 reports the average learning accuracy (the accuracy on a task immediately after training on that task) and the knowledge accumulation on a downstream task. For all methods, relative forgetting in the representation is at least as bad as at the output level, except for the first task. Relative forgetting in the representation for ER and MAS is lower than for finetuning, yet their knowledge accumulation is similar. Adding more samples to the memory in ER does improve knowledge accumulation and reduces overall forgetting, see Appendix. One explanation for this is their lower learning accuracy; they learn less in the first place. LwF has even lower relative forgetting, and in contrast to ER

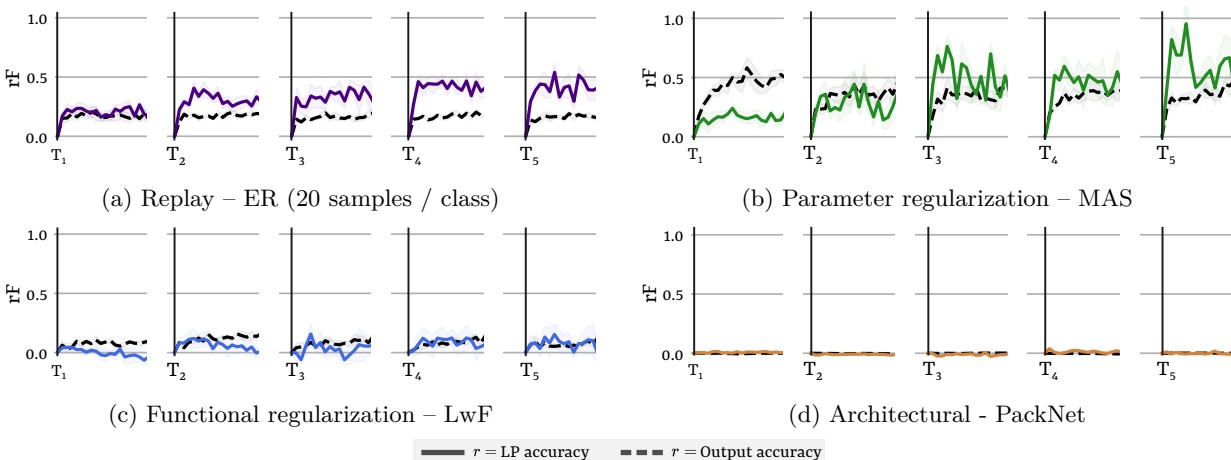

(a) Replay – ER (20 samples / class)  (b) Parameter regularization – MAS

(c) Functional regularization – LwF  (d) Architectural - PackNet

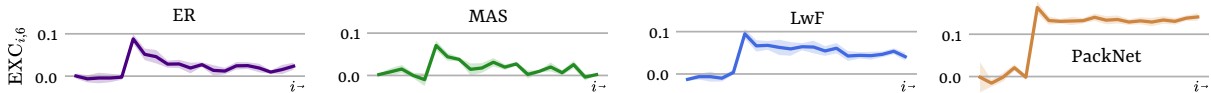

Figure 6: Relative Forgetting at the output level and the level of representation as in Eq. 1, for $T_1$ to $T_5$ of the tested methods on the Mini-ImageNet sequence. (Mean $\pm$ SE, 5 runs)

Figure 7: Task exclusion results of the tested methods on $T_6$ of the Mini-ImageNet sequence. Results with $EXC_{i,j} > 0$ show that some task-specific information is retained. (Mean $\pm$ SE, 5 runs)

and MAS, the exclusion baseline shows that it remembers at least some information specific to the trained task. Its knowledge accumulation is between the ensemble and finetuning, indicating that more task-specific information is retained than with the other methods, but not everything. PackNet is an entirely different story. It does not forget, but what it learns is a lot less useful for other tasks. This is likely because by having masks specific to a single task, the representation for one task is nothing like the representation for another, hindering effective learning and knowledge accumulation.

Table 2: Learning accuracy averaged over all tasks ($\overline{\mathrm{LP}_{i,i}}$) and knowledge accumulation on a downstream task ($\mathrm{LP}_{19,d} - \mathrm{LP}_{0,d}$), reported for the two baselines in Section 4.3 and the methods of Section 5. (Mean $\pm$ SE, 5 runs)

|  | Finetune | Ensemble | ER | MAS | LwF | Packnet |
|---|---|---|---|---|---|---|
| $\overline{\mathrm{LP}_{i,i}}$ | $0.860 \pm 0.002$ | $0.864 \pm 0.002$ | $0.815 \pm 0.002$ | $0.792 \pm 0.001$ | $0.852 \pm 0.001$ | $0.792 \pm 0.003$ |
| $\mathrm{LP}_{19,d} - \mathrm{LP}_{0,d}$ | $0.249 \pm 0.023$ | $0.365 \pm 0.020$ | $0.247 \pm 0.018$ | $0.259 \pm 0.013$ | $0.321 \pm 0.019$ | $0.195 \pm 0.022$ |

## 6 Discussion and future work

This paper sheds light on the issues of feature forgetting and knowledge accumulation and can guide future research in developing new and better methods to learn representations continually. Representations of one task can help other tasks, so it might not be necessary to train later tasks as if it was the first task. Part of the information will already be learned by the model, so the model should not need to relearn this. Similarly, not every sample might be equally important to be part of a replay memory to maintain high performance. Some of the samples can also be learned from other tasks and thus may not need to be replayed. To improve knowledge accumulation, functional regularization seems a promising candidate, and is thus recommended to be explored further. Understanding why it works relatively well and how it can be improved upon are both interesting directions for future work.

It is possible to think of two tasks that have nothing in common and thus learning one task does not improve the representation for the other. Yet, tasks composed of natural images, like those in the benchmarks

we used, do share information. An open question is how much of the shared information is at the image level (*e.g.* detecting edges and shapes) and how much at the semantic level (*e.g.* being able to recognize a specific feature, like eyes or car tyres). Further unraveling these two might lead to important insights in how neural networks operate. Likewise, in this paper we always used $n$-way classification in both training and testing, with $n$ constant throughout each experiment. The influence of changing problems during training or evaluation (*e.g.* a different amount of classes per task during evaluation than during training) on the representation quality is something that should be considered in future work.

## 7 Background

In this section we provide more background information related to the results in this paper. They are not necessary to understand our paper, but they may be good starting points to explore the subject further.

**Representation learning**. Data rarely come in a format that is adapted to the task we want to perform (Bengio et al., 2013). Except for very simple problems, it is near impossible to directly classify images in their raw pixel representation. For a long time, researchers have been searching for a representation of images that makes it convenient to solve semantic tasks. Handcrafting features was the standard (*e.g.* Csurka et al., 2004), but this requires expert knowledge engineering and may not result in optimal features. Since the rise of deep learning, features are more commonly learned by neural networks, directly from the raw data. Both Bengio et al. (2013) and Goodfellow et al. (2016) define *good* representations as ones that make it easier to solve tasks of interest, a definition we adopt. They see deep neural networks as inevitable representation learners, even when this is not explicitly the goal. Neural networks trained to predict image-label pairs indirectly learn a representation where semantically different images are linearly separable in the output of the penultimate layer. Yet representations can also be learned directly, which can improve robustness, boost generalization, or reduce the need for labeled data (Jing & Tian, 2020).

**Head vs. representation**. The paper proposing iCaRL (Rebuffi et al., 2017) is one of the first continual learning works to explicitly disentangle the representation and head. The head of a model can be relatively well learned with small subsets of data, *e.g.* in the case of classification as a linear layer or with non-parametric approaches like $k$-nearest neighbors (Wang et al., 2020; Taunk et al., 2019). On the other hand, heads often do not transfer well, and can quickly become disconnected from the representation when the representation changes while the head is static (Caccia et al., 2021). In the context of continual learning this property has been identified to impact performance severely, and methods updating the last layer only on small memories with balanced data, have shown successes in overcoming much of the observed forgetting (Wu et al., 2019; Zhao et al., 2020).

Recently some continual learning methods explicitly try to foster transfer of knowledge by taking inspiration from advances in representation learning (Jing & Tian, 2020). Some approaches apply contrastive losses (Cha et al., 2021; Mai et al., 2021) and self-supervised learning (Marsocci & Scardapane, 2023; Hu et al., 2022; Fini et al., 2022; Rao et al., 2019) to improve continual learning performance, other works take ideas from meta-learning (Javed & White, 2019; Caccia et al., 2020) to learn representations that can easily adapt to new tasks. Lastly, Pham et al. (2021) take inspiration from neuroscience and combine fast and slow learners, *i.e.* supervised and self-supervised modules, in one system.

**Evaluating representation quality**. Effectively leveraging generalization and transfer properties of deep representations is one thing, evaluating their quality is another. As pointed out throughout this work, measuring forgetting at the output (the head) of a neural network does not tell us everything about the internal state of a network. Studies that retrain the last layer (Xiong et al., 2019), or a set of deeper layers (Murata et al., 2020), with the earlier layers frozen, hint that representations of lower layers are still useful for seemingly forgotten tasks. However, rather than these layers remembering something specific to the observed tasks, other works interpret this as better generalizability of the lower layers (Ramasesh et al., 2021; Yosinski et al., 2014; Zeiler & Fergus, 2014). Early layers may not seem to forget as much, because their representations are so general that they are almost fully reusable for future tasks, while deeper layers successively encode information more specific to the observed data, that is prone to being overwritten by information of new task's data (Ramasesh et al., 2021).

A number of recent works use linear probing (Alain & Bengio, 2016) to analyze continually learned representations at the penultimate layer (Hu et al., 2022; Davari et al., 2022; Zhang et al., 2022; Chen et al., 2023; Kim & Han, 2023). Considering the amount of forgetting in the continually learning representations, Davari et al. (2022) conclude that this forgetting is less catastrophic than at the output and suggest that no task-critical information is lost. Hu et al. (2022) and Zhang et al. (2022) evaluate the representation's quality with respect to downstream tasks and arrive at similar findings as Davari et al.. Yoon et al. (2023) consider a continual pre-training setting that allows fine-tuning of all parameters in the model during probing. They report incrementally improving performance on a downstream task, and ascribe it to an increasing transferability of the features learned continually. In contrast to these studies just listed, Kim & Han (2023) show severe forgetting in the representation, but start from models pre-trained on half the respective dataset rather than the more even split considered in Davari et al. (2022). Chen et al. (2023) show that continual learners that have less forgetting at the representation level are better few-shot learners, hinting at the same relation we show with our ensemble baseline.

## 8 Conclusion

In this work we studied how deep neural networks learn and forget representations when continually trained on a sequence of image classification tasks. If forgetting is expressed as the proportion of newly gained performance that is forgotten, representations forget about as much as at the output level, except for the first tasks. During the first task, information that is shared across tasks is learned for the first time. Because this shared information is repeated to the model in each task, it is not forgotten. The information that is forgotten at the representation level reduces how much knowledge a model accumulates, as exemplified by the *ensemble* baseline. Finally, we compared the feature forgetting and knowledge accumulation of different types of continual learning methods, whereby we found that functional regularization can prevent a large portion of representation forgetting. We hope that the insights provided by our work inspire the development of continual learning methods with less feature forgetting and more knowledge accumulation.

### Broader Impact Statement

Successful continual learning can reduce the need for the ever larger growing compute required, without the need for retraining from scratch, reducing the environmental costs. It would allow fixing mistakes that machine learning models make in a more efficient way and hence contribute to machine learning safety. As with all machine learning applications, there always is a dual-use risk. However, given the exploratory nature of this work, we believe the risk of this particular work to be limited.

### Acknowledgments

This paper is part of a project that has received funding from the European Union under the Horizon 2020 research and innovation program (ERC project KeepOnLearning, grant agreement No. 101021347) and under Horizon Europe (Marie Skłodowska-Curie fellowship, grant agreement No. 101067759).

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

## Appendix

The Appendix contains additional information for the content presented in the main body. The first two sections provide further detail of our experimentation settings and the ensemble baseline. Thereafter, we report on additional experiments: First, regarding continual representation learning with supervised contrastive and a self-supervised training approaches, and second, starting from a higher quality representation obtained pre-trained weights, rather than training from-scratch. Further we provide the full plots for relative forgetting that were not used in the main body due to spatial constraints, re-evaluation of all our experimentation on the CIFAR-100 dataset, as well as consideration of an alternative probing mechanism to quantify representation quality, namely $k$-NN. Finally, we list detailed task sequence information to improve reproducability of our results. In summary, the the supplementary material is ordered as follows:

## A  Experimentation details

This section details the training and evaluation of all experiments in the main paper and supplemental material, unless explicitly stated to deviate.

**Data**  MiniImageNet consists of $50,000$ train and $10,000$ test RGB-images of resolution $84 \times 84$ equally divided over 100 classes. We split this dataset into 20 disjoint tasks such that each task contains five classes. The second benchmark is Split CIFAR-100, which is based on the CIFAR-100 dataset (Krizhevsky et al., 2009) with the same amount of RGB-images and classes as MiniImageNet, but with reduced resolution of $32 \times 32$. We split this dataset into ten disjoint tasks with ten classes each. All experiments are run with five different seeds that also shuffle the class splits over the tasks. See Table 4 and Table 5 for the exact sequences.

**Architecture and optimization**  Throughout this work ResNet-18 (He et al., 2016) is the base architecture for all models. For MiniImageNet we adopt the implementation as default in the pytorch-torchvision (Paszke et al., 2019) library. For CIFAR-100 we employed the slim version of the model as proposed by Lopez-Paz & Ranzato (2017). All networks are trained from scratch, if not explicitly specified otherwise. The optimization schedules are adjusted with respect to the training criterion. For supervised training with the cross-entropy loss we use an AdamW (Loshchilov & Hutter, 2017) optimizer with static learning rate of 0.001, weight decay 0.0005, and beta-values 0.9 and 0.999. Each task is trained for 50 epochs with mini-batches of size 128.

For the SupCon (Khosla et al., 2020) and Barlow Twins (Zbontar et al., 2021) optimization criteria, we stuck to optimization schedules proposed in literature for their application to continual learning. In line with observations by Cha et al. (2021), the SupCon training regime uses an SGD optimizer with momentum 0.9. The learning rate is scheduled in the same way for every task warming up from 0.0005 to 0.1 in the first ten epochs, then annealing by a cosine schedule back to its starting value. The first task is trained for

500 epochs, all subsequent tasks for 100 epochs, with a batch size of 256. The projection network necessary for this objective consists of an MLP with (single) hidden dimension of 512, projecting to a 128 dimensional space. Barlow-Twins optimization is aligned to (Marsocci & Scardapane, 2023; Fini et al., 2022). We use an Adam optimizer (Kingma & Ba, 2015) with learning rate 0.0001 and weight decay 0.0005. We train 500 epochs for each task with batch size of 256. Again, the projection head is an MLP but with two hidden layers, and hidden and final projection dimension of 2048. All methods use the same augmentations, see below.

**Probe optimization** To quantify the quality of the representation we apply probes based on linear classifiers and $k$-nearest neighbors ($k$NN). Linear classifiers consist of a single linear layer and are optimized with access to all training data, in a way analog to Cha et al. (2021). Keeping a batch-size of 128, we use SGD with momentum of 0.9 and no weight decay for 100 epochs. The learning rate of 0.1 is decaying at epochs 60, 75, and 90 by a factor of 0.2. Similarly, $k$NN uses all training data to evaluate the representations, with $k = 20$.

**Continual learning mechanisms** LwF and MAS are using a value of $\lambda = 1.0$ as advocated by its original authors. Replay uses a random selection of 20 exemplars per class. The weight of the loss on replayed samples is increased proportionally to the number of previously observed tasks, to prevent favoring the current task in the optimization. PackNet prunes 75% of the model's parameters in each layer, followed by a post-pruning phase of 25 epochs ($1.5\times$ training epochs), similar to the settings reported in its original work. An upper bound is reported by jointly training the model on all observed data. For our lower-bound we want to document the impact the singled out tasks have. This we achieve by re-initializing the model before training a new task, but allowing the new task to train for as many iterations as a continual model would have, *e.g.* 50 epochs for the first task, then 100 for the second, and so on. By design this model has zero transfer of knowledge, and we will refer to it as 'Single task' baseline.

**Augmentations** In all experiments we use the data augmentation pipeline from SimCLR (Chen et al., 2020). The augmentations pipeline consists of random crops and horizontal flips, color-jitter (brightness=0.4, contrast=0.4, saturation=0.2, hue=0.1), random grayscaling (p=20%) and Gaussian blur using a kernel of size 9 and sigma range 0.1 to 0.2. In PyTorch, the augmentations are defined as follows:

```
from torchvision.transforms import *

RandomHorizontalFlip(p=0.5),
RandomResizedCrop(size=(32, 32), scale=(0.2, 1.0)),
RandomApply(
    [ColorJitter(brightness=0.4, contrast=0.4, saturation=0.2, hue=0.1)], p=0.8),
RandomGrayscale(p=0.2),
RandomApply([
    GaussianBlur(kernel_size=input_size[0]//20*2+1, sigma=(0.1, 2.0))], p=0.5)
```

## B  Ensemble: further details

The ensemble method stores a model copy after every task's training. Each of these models output a representation $f_i$ with dimension $k$ for the input data. During training, only the model of the current, $i$-th, task is used and the others are frozen. During training, the model learns exactly in the same way as the default continually fine-tuned model (*c.f.* Section 2). For example, when training on the $n$-th task, the $n$-th sub-model of the ensemble baseline (*i.e.* the sub-model being trained on the $n$-th task), is precisely the same as the fine-tuned model when it is being trained on the $n$-th task. The difference is in the evaluation. During evaluation, not only the latest model's representation $f_t$ is used, but all of the representations are concatenated to form one large representation $f = [f_1, f_2 \cdots f_t]$. Note that because learning was exactly the same, all "intermediate models" $\{f_1, ..., f_{t-1}\}$ do not contain any information that was not at some point also learned by the fine-tuned model. Hence, the ensemble model does not use any additional information; it is keeping track of information that is forgotten by the fine-tuned model. On top of this large representation $f$, a linear layer with input dimension $tk$ is trained, instead of just $k$ for the finetuned model.

To mitigate the influence of the higher dimension, for which it might be easier to find linearly separable features, we add a dimension reduction to lower the dimension back to $k$. We do this by projecting the features of the ensemble on the top-$k$ most significant PCA dimensions. The results are shown in Figure 8. The reduced ensemble performs slightly worse than the full ensemble, yet still significantly better than finetuning.

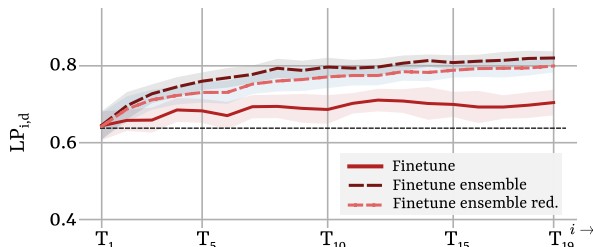

Figure 8: LP-accuracies of finetuning, the ensemble baseline and its reduced version, as explain in Section B

## C  Supervised contrastive and self-supervised Barlow Twins

In this section, we extent our investigation of the apparent innate robustness to forgetting for continually learned representations from supervised fine-tuning with the cross-entropy (CE) loss (*c.f.* Section 4), to the alternative training approaches of contrastive learning and self-supervised learning. The recent success (Ericsson et al., 2022) of these learning mechanisms sparked interest for their application in continual learning as well (Fini et al., 2022; Cha et al., 2021; Marsocci & Scardapane, 2023; Wang et al., 2023), and they have been found to be less prone to forgetting Wang et al. (2023), which got linked to the improved generality of the learned representation (Tendle & Hasan, 2021). From the plethora of potential methods, we chose SupCon (Khosla et al., 2020) and Barlow Twins (Zbontar et al., 2021) as representatives of contrastive and self-supervised losses. Both have already been used in continual learning settings (Marsocci & Scardapane, 2023; Cha et al., 2021; Davari et al., 2022; Cha et al., 2022). In the following, we repeat to quantify the quality of a learned representation by its linear probing (LP) accuracy on the tasks of the continual training sequence $\mathcal{T}$, as well as a downstream task $T_d$ that is never seen during training. First, we provide an intuition to the continual learning performance of SupCon and Barlow Twins in contrast to supervised CE fine-tuning. Then, we analyze forgetting analog to the main text by relative forgetting (*c.f.* Section 4.1) and task exclusion difference (*c.f.* Section 4.2). Finally, we again link the effect of feature forgetting to reduced knowledge accumulation efficiency using the ensemble baseline (*c.f.* Section 4.3), before summarizing the results.

**Learning and forgetting SupCon and Barlow Twins representations**  To provide a frame-of-reference we first relate the linear probing (LP) accuracies of continual SupCon, Barlow Twins, and fine-tuning with supervised CE loss in Figure 9. This qualitative comparison provides intuition to their evolving representation quality on the training sequence and downstream task. Continual SupCon almost resembles the continual supervised CE fine-tuning baseline. It displays prominent peaks and decay after, as well as some knowledge accumulation for the downstream task. However, in our experiments continual SupCon falls short of the performance obtained from CE fine-tuning. With continual Barlow Twins, the increase in performance from training and evaluating on the same task $t$ ($LP_{t,t}$) is comparatively limited. Yet, the obtained representations are more general regarding other tasks in the sequence and show increased knowledge accumulation.

We introduced relative forgetting in Section 4.1 of the main body as an evaluation perspective that reveals how forgetting at the output level and representation level is equally catastrophic once we consider the knowledge already contained in our continual learning model. For SupCon and Barlow Twins a direct comparison is not possible, since there is no output performance to be measured. Both require a second (head-) training stage, to apply the learned representation to a task by annotated data. Still, the relative forgetting plots in figure 10 offer a second perspective to theevolution of representation quality in figure 9.

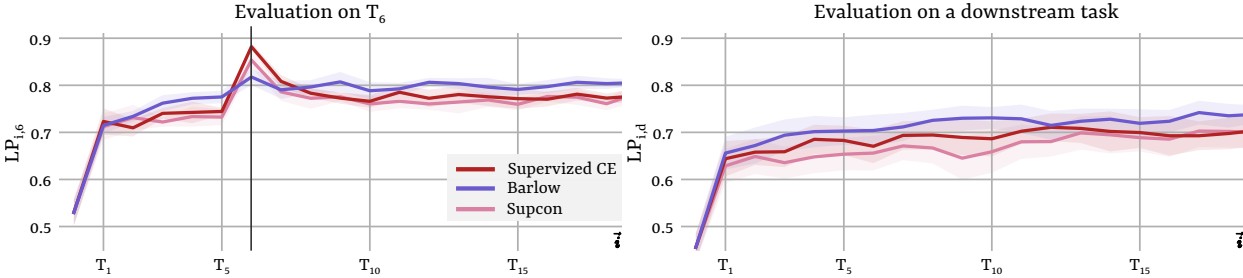

Figure 9: Illustration of feature forgetting and knowledge accumulation for continually finetuned SupCon and Barlow Twins on Split MiniImageNet, in comparison to continual supervised finetuning using the Cross-Entropy loss (*i.e.* finetuning in the main paper).

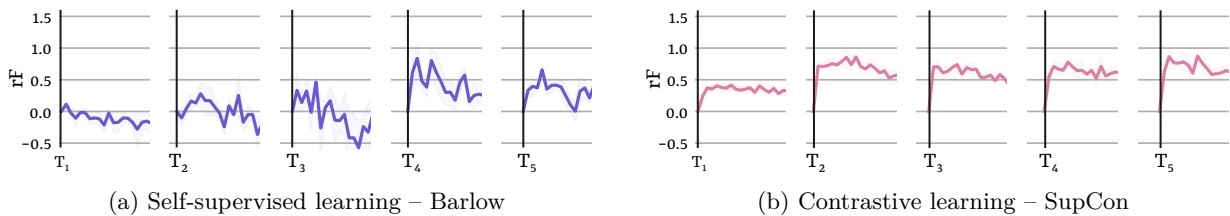

(a) Self-supervised learning – Barlow        (b) Contrastive learning – SupCon

Figure 10: Forgetting for the first 5 tasks using self-supervised and contrastive losses, instead of the default cross-entropy. Output accuracy is not properly defined for these methods, they always have to train a final linear layer. (Mean $\pm$ standard error)

SupCon rapidly forgets newly learned information from each task in continued training, but still achieves a certain degree of knowledge accumulation. Barlow Twins demonstrates more unstable relative forgetting, but ultimately shows the same trend. The information gain for any task is not as pronounced as when training with additional labels provided for the data. Despite their increased transferability, features can become quickly forgotten to the extent that the accumulating knowledge surpasses early training performance for initial tasks.

Also the task exclusion difference ($EXC_{i,j}$, *c.f.* Eq. 2), depicted in figure 11, confirms the previous observations. The task exclusion measure allows to study the information gain with respect to a particular task, while accounting for the knowledge accumulation with respect to all other tasks in the training sequence. For both, SupCon and Barlow Twins, the information gain from any individual tasks is marginal. At the end of continual training the gap is on average reduced to almost zero.

**Feature forgetting reduces knowledge accumulation**    Lastly, to study the impact of the observed feature forgetting to knowledge accumulation, we re-employ the ensemble baseline introduced in Section 4.3. It allows us to quantify the knowledge accumulation without the impact of forgetting throughout continual training. From the comparison between continually fine-tuning and its ensemble counterpart in Figure 12, we can find an ample increase in the quality of the representation for SupCon and Barlow Twins when using

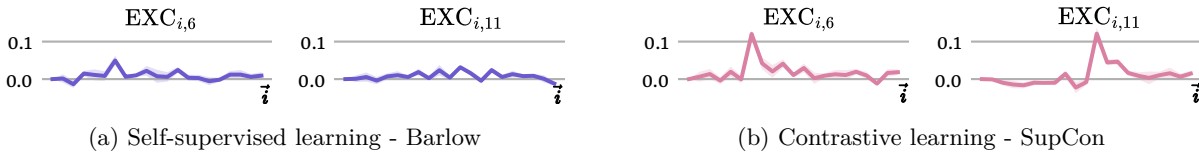

(a) Self-supervised learning - Barlow        (b) Contrastive learning - SupCon

Figure 11: Task exclusion difference for continually fine-tuned SupCon and Barlow Twins on Split MiniImageNet for tasks $T_6$ and $T_{11}$

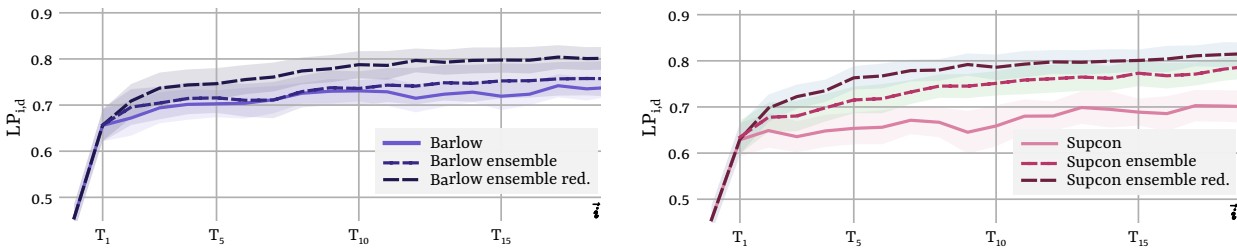

Figure 12: Ensemble baseline comparison for SupCon and Barlow Twins. Ensemble referes to the full ensemble, 'ensemble red.' is the ensemble that has its dimension reduced again by PCA.

the ensemble. This clearly shows the impeding effect of feature forgetting on the representation learning also for supervised contrastive and self-supervised learning. We note that Barlow Twins does suffer a lot more from correcting for the ensemble's dimensionality increased through PCA, compared to the supervised approaches. An explanation to this might be in the definition of the Barlow Twins' loss (Zbontar et al., 2021). Because it explicitly optimizes towards a high rank representation, each task's representation should spread a high number of significant components. Assuming a considerable quantity of information is not shared between tasks, as a result, reducing the concatenated representations by keeping its most significant components does not maintain all tasks' information. Supervised approaches suffer a lot less in this regard since they typically employ few significant components (Feng et al., 2022) in the representation space. This leaves enough effective encoding space for information from all tasks.

**Do they forget catastrophically?** In summary, we confirm similar trends for relative forgetting and task exclusion difference with SupCon and Barlow Twins as we detailed for continual supervised CE fine-tuning in our main results. Also, using the ensemble baseline, we demonstrate that feature forgetting hinders efficient continual representation learning for these methods as well. However, we choose not to claim a definite conclusion on whether forgetting should be considered catastrophic or not. This is for the reason that we cannot link the common expression of catastrophic forgetting of supervised CE settings to SupCon or Barlow Btwins directly, as we did in the main body, due to the there being no meaningful output accuracy measure. We emphasize that the results listed in this section suggest that, forgetting is indifferent to the training loss and supervision. What does change is the "kind" of features extracted from the data by the specific learning mechanism - approaches such as Barlow Twins may render features to be more similar across tasks. We exemplified how this is giving the impression of less forgetting, but can similarly be explained by the repeated re-exposure of the model to the specific feature(s).

## D   Pre-trained representations

Similarly to representations being attributed an innate robustness to forgetting when learning continually, pre-trained backbones receive this reputation as well (Lee et al., 2023; Zhang et al., 2023). In light of our study, here we also consider the extent to which representations considered "high quality", such as obtained from large-scale pre-training (Kornblith et al., 2019), impact forgetting at representation level. For this experiment we simply exchange the random initialization that we have been using through the main body with weights from a backbone pre-trained on ImageNet1k  (Deng et al., 2009) [2]. An innate robustness to forgetting of the representation should manifest as overall improved performance due to the increased generality and/or quality of the pre-trained features. Note that we exemplarily investigate an extreme case where pre-training (ImageNet1k) coveres an almost identical data distribution as the continual training sequence (Split Mini-Imagenet).

Little surprisingly, the results in Figure 13 show that, for our case of Split of Mini-Imagenet, the representation of the pre-trained backbone ($T_0$) is already a good foundation for the task sequence. Nevertheless, with sequential training the initial generality is quickly lost and we again observe the training dynamics governed

---

[2]weights are publicly available in the PyTorch-Torchvision library

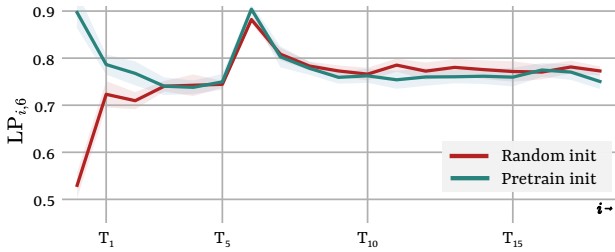

Figure 13: Comparison of continual learning linear probing (LP) when training from-scratch (random weight initialization) or pre-trained weights (ImageNet-1k).

by feature forgetting and knowledge accumulation as we describe them in the main body (*c.f.* Section 4). We hypothesize that, even though the features obtained from ImageNet pre-training are well suited for the sequential training task, they cannot be similarly obtained from any of the tasks in the continual training sequence. As a results, they are forgotten *despite* their generality. Our finding aligns with the work of Kim & Han (2023), who show that continual learning approaches making effective use of pre-trained representations either drastically reduce their learner's plasticity or otherwise extend the initial representation without altering it, to circumvent forgetting.

## E  Relative forgetting: more replay results

In Figure 14 we show relative forgetting on the Mini-ImageNet sequence when storing 20, 50 or 100 samples per class, and when adding distillation using the output logits (DER (Buzzega et al., 2020)). Figure 15 shows the exclusion results of $T_6$ and Table 3 the amount of knowledge accumulation. Having more samples in the memory reduces both forgetting in the representation and the head and substantially improves knowledge accumulation. Interestingly, the final exclusion difference remains small. This supports our theory. When adding more replay samples the model can learn more from the combination of replay and new task data, but what cannot be learned from this combination of data is still lost. DER is mostly helpful to further reduce forgetting in the head. When storing the output logits for each sample, more information is stored, and it is possible that this reduces the risk of overfitting of the head on the few samples that are in the memory.

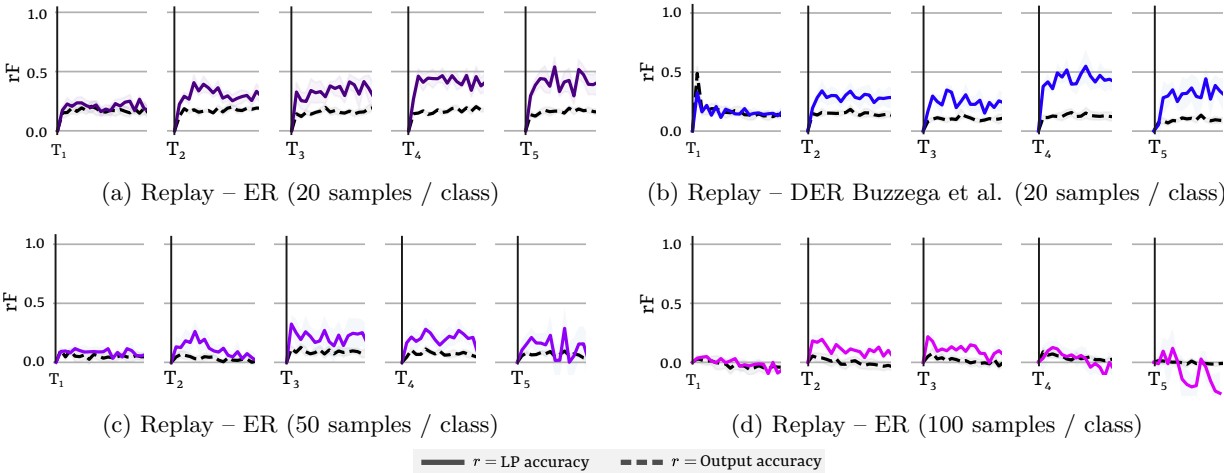

Figure 14: Relative Forgetting at the output level and the level of representation as in Eq. 1, for $T_1$ to $T_5$ of more replay variants on the Mini-ImageNet sequence. The influence of storing more samples and adding distillation with soft logits (DER (Buzzega et al., 2020)) is evaluated. (Mean $\pm$ SE, 5 runs)

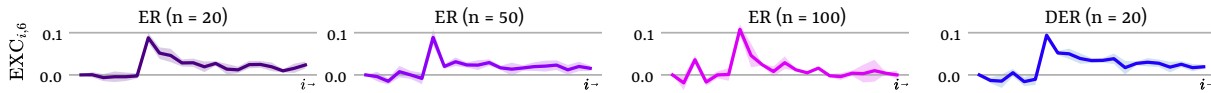

Figure 15: Task exclusion results of more replay variants on the Mini-ImageNet sequence. The influence of storing more samples and adding distillation with soft logits (DER (Buzzega et al., 2020)) is evaluated. (Mean $\pm$ SE, 5 runs)

Table 3: Learning accuracy averaged over all tasks ($\overline{\mathrm{LP}_{i,i}}$) and knowledge accumulation on a downstream task ($\mathrm{LP}_{19,d} - \mathrm{LP}_{0,d}$), reported when adding more replay samples and with DER (Buzzega et al., 2020). (Mean $\pm$ SE, 5 runs)

| | Finetune | Ensemble | ER ($n = 20$) | ER ($n = 50$) | ER ($n = 100$) | DER ($n = 20$) |
|---|---|---|---|---|---|---|
| $\overline{\mathrm{LP}_{i,i}}$ | $0.860 \pm 0.002$ | $0.864 \pm 0.002$ | $0.815 \pm 0.002$ | $0.814 \pm 0.003$ | $0.821 \pm 0.003$ | $0.826 \pm 0.003$ |
| $\mathrm{LP}_{19,d} - \mathrm{LP}_{0,d}$ | $0.249 \pm 0.023$ | $0.365 \pm 0.020$ | $0.247 \pm 0.018$ | $0.298 \pm 0.014$ | $0.309 \pm 0.015$ | $0.271 \pm 0.013$ |

# F    Relative forgetting: extra results

Figure 3 only shows the relative forgetting in the Mini-ImageNet task sequence. For completeness, we report here the relative forgetting of all tasks in Figure 16.

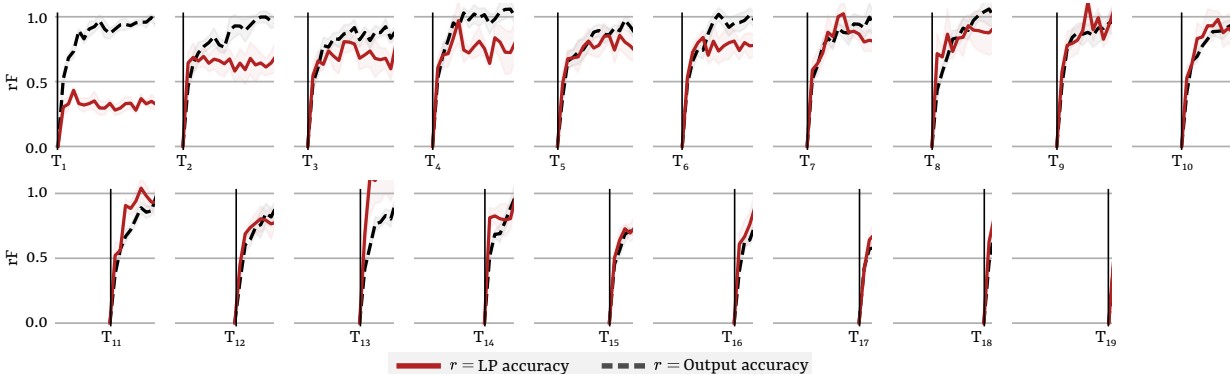

Figure 16: Representation and observed forgetting using linear probes for all tasks in Mini-ImageNet using finetuning (except first and last, for which we cannot calculate relative forgetting)

## G   Results on CIFAR-100

To reduce the dependency on only having experiments on a single dataset, we report our main results here also on CIFAR-100. The results on CIFAR-100 follow the same general trends as those on Mini-ImageNet in the main paper. The largest difference is that the effects are sometimes smaller, due to the shorter task sequence.

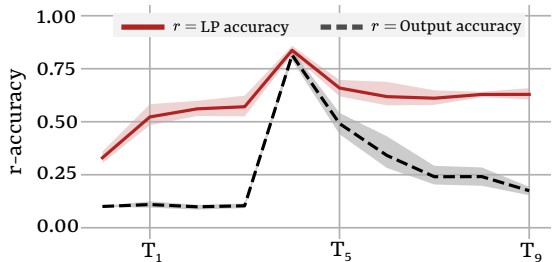

Figure 17: Linear probe and output accuracy of $T_4$ during the entire CIFAR-100 sequence.

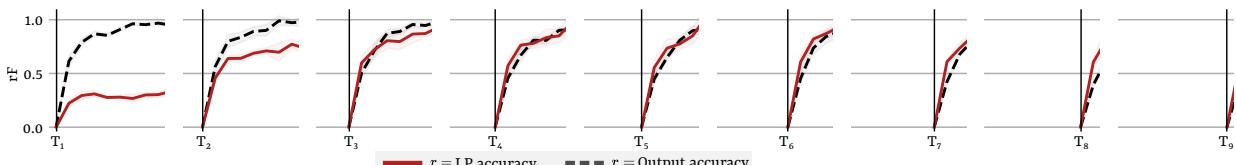

Figure 18: Representation and observed forgetting using linear probes for all tasks in CIFAR-100 using finetuning (except first and last task, for which we cannot calculate relative forgetting)

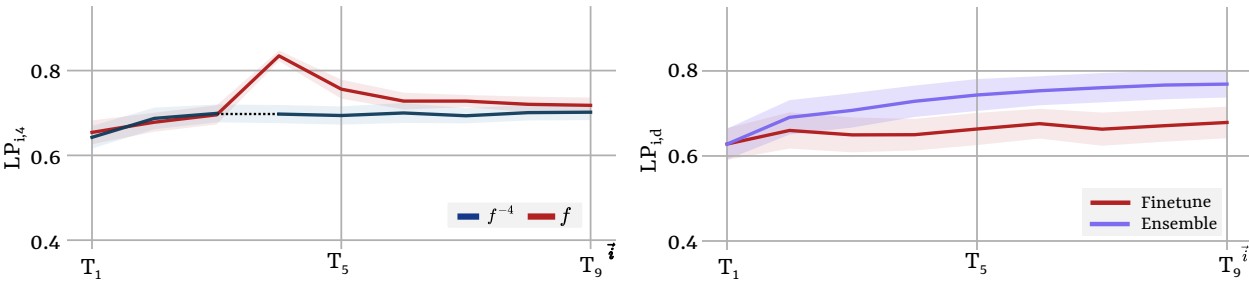

Figure 19: Finetune, exclusion, single task and multi task with CIFAR-100.

Figure 20: Comparing the ensemble and finetuning on CIFAR-100.

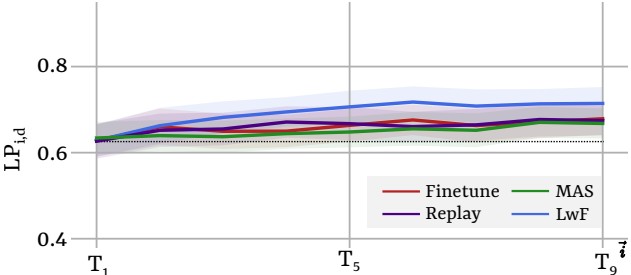

Figure 21: LP-accuracies on a downstream task of CIFAR-100.

## H    Evaluation with *k*NN

In this section we report the most important results of the main paper using *k*NN instead of using linear probes. This has the benefit that it there are no hyper-parameters to tune and does not depend on the optimization used. We report it here for completeness, and keep the linear probes in the main paper as this is how previous papers reported their results Davari et al. (2022); Cha et al. (2022); Zhang et al. (2022). In general, the results in Figure 22, 23, 24 and 25 follow the same trends as observed in the main paper, with the main difference that the absolute values are lower, likely due to the suboptimalitiy of *k*NN compared to linear probe optimization.

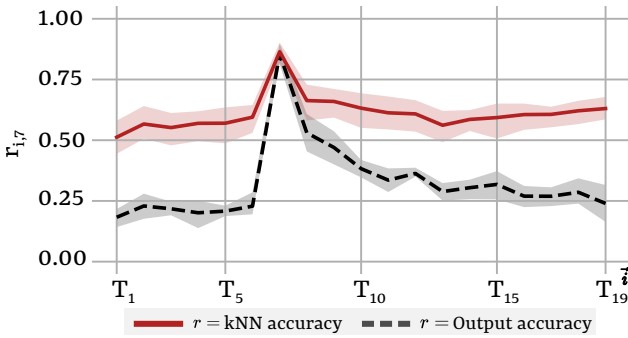

Figure 22: *k*NN and output accuracy of $T_7$ during the entire Mini-ImageNet sequence.

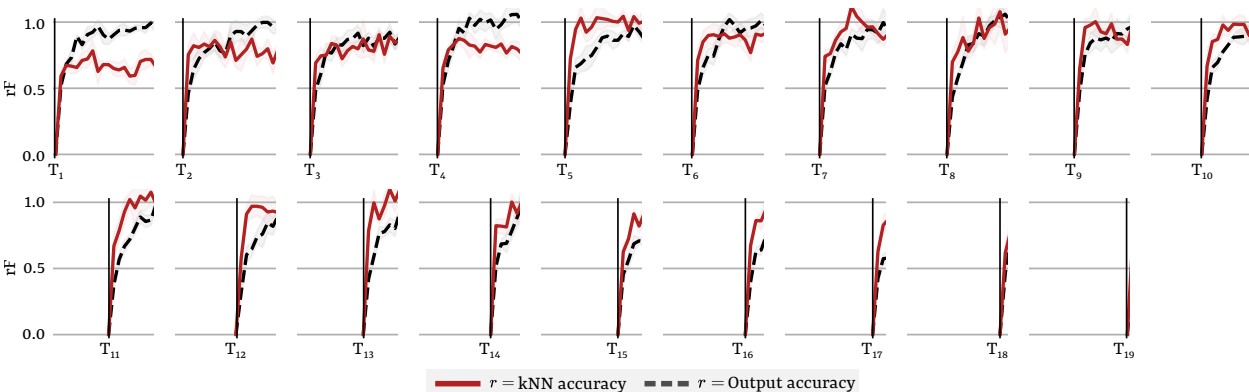

Figure 23: Representation and observed forgetting using *k*NN for all tasks in Mini-ImageNet (except last and first, for which we cannot calculate relative forgetting)

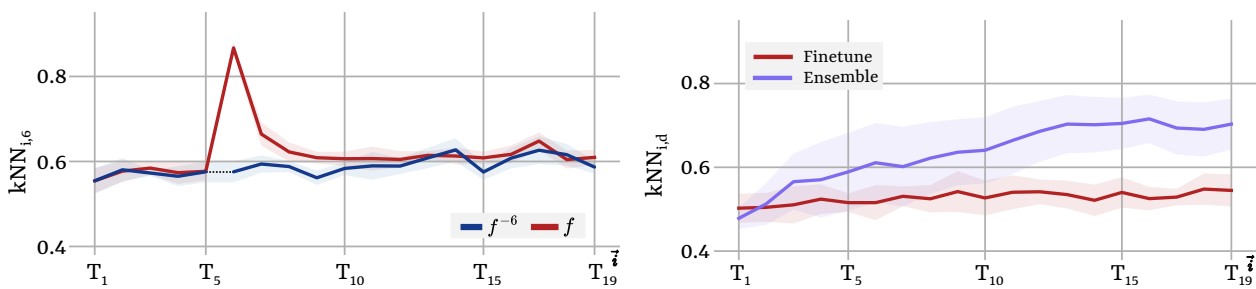

Figure 24: Finetune, exclusion, single task and multi task using *k*NN.

Figure 25: Comparing the ensemble and finetuning using *k*NN.

# I   Detailed task sequence information

In Table 4 and Table 5 we report the exact task sequences used in the experiments in the main paper. These are chosen at random, but consistent in all experiments. The randomness of the tasks also means that there difficult varies quite a bit, which explais some of the higher standard errors in the experiments reported.

| idx | Synset | Synset name | idx | Synset | Synset name | idx | Synset | Synset name |
|---|---|---|---|---|---|---|---|---|
| 0 | n01532829 | house_finch | 33 | n03400231 | frying_pan | 66 | n02981792 | catamaran |
| 1 | n01558993 | robin | 34 | n03476684 | hair_slide | 67 | n03980874 | poncho |
| 2 | n01704323 | triceratops | 35 | n03527444 | holster | 68 | n03770439 | miniskirt |
| 3 | n01749939 | green_mamba | 36 | n03676483 | lipstick | 69 | n02091244 | Ibizan_hound |
| 4 | n01770081 | harvestman | 37 | n03838899 | oboe | 70 | n02114548 | white_wolf |
| 5 | n01843383 | toucan | 38 | n03854065 | organ | 71 | n02174001 | rhinoceros_beetle |
| 6 | n01910747 | jellyfish | 39 | n03888605 | parallel_bars | 72 | n03417042 | garbage_truck |
| 7 | n02074367 | dugong | 40 | n03908618 | pencil_box | 73 | n02971356 | carton |
| 8 | n02089867 | Walker_hound | 41 | n03924679 | photocopier | 74 | n03584254 | iPod |
| 9 | n02091831 | Saluki | 42 | n03998194 | prayer_rug | 75 | n02138441 | meerkat |
| 10 | n02101006 | Gordon_setter | 43 | n04067472 | reel | 76 | n03773504 | missile |
| 11 | n02105505 | komondor | 44 | n04243546 | slot | 77 | n02950826 | cannon |
| 12 | n02108089 | boxer | 45 | n04251144 | snorkel | 78 | n01855672 | goose |
| 13 | n02108551 | Tibetan_mastiff | 46 | n04258138 | solar_dish | 79 | n09256479 | coral_reef |
| 14 | n02108915 | French_bulldog | 47 | n04275548 | spider_web | 80 | n02110341 | dalmatian |
| 15 | n02111277 | Newfoundland | 48 | n04296562 | stage | 81 | n01930112 | nematode |
| 16 | n02113712 | miniature_poodle | 49 | n04389033 | tank | 82 | n02219486 | ant |
| 17 | n02120079 | Arctic_fox | 50 | n04435653 | tile_roof | 83 | n02443484 | black-footed_ferret |
| 18 | n02165456 | ladybug | 51 | n04443257 | tobacco_shop | 84 | n01981276 | king_crab |
| 19 | n02457408 | three-toed_sloth | 52 | n04509417 | unicycle | 85 | n02129165 | lion |
| 20 | n02606052 | rock_beauty | 53 | n04515003 | upright | 86 | n04522168 | vase |
| 21 | n02687172 | aircraft_carrier | 54 | n04596742 | wok | 87 | n02099601 | golden_retriever |
| 22 | n02747177 | ashcan | 55 | n04604644 | worm_fence | 88 | n03775546 | mixing_bowl |
| 23 | n02795169 | barrel | 56 | n04612504 | yawl | 89 | n02110063 | malamute |
| 24 | n02823428 | beer_bottle | 57 | n06794110 | street_sign | 90 | n02116738 | African_hunting_dog |
| 25 | n02966193 | carousel | 58 | n07584110 | consomme | 91 | n03146219 | cuirass |
| 26 | n03017168 | chime | 59 | n07697537 | hotdog | 92 | n02871525 | bookshop |
| 27 | n03047690 | clog | 60 | n07747607 | orange | 93 | n03127925 | crate |
| 28 | n03062245 | cocktail_shaker | 61 | n09246464 | cliff | 94 | n03544143 | hourglass |
| 29 | n03207743 | dishrag | 62 | n13054560 | bolete | 95 | n03272010 | electric_guitar |
| 30 | n03220513 | dome | 63 | n13133613 | ear | 96 | n07613480 | trifle |
| 31 | n03337140 | file | 64 | n03535780 | horizontal_bar | 97 | n04146614 | school_bus |
| 32 | n03347037 | fire_screen | 65 | n03075370 | combination_lock | 98 | n04418357 | theater_curtain |

Table 4: The classes included in Split MiniImagenet, with their index, (which is not general, but used in the task splits), their synsets and their name.

| | Seed 42 | Seed 52 | Seed 62 | Seed 72 | Seed 82 |
|---|---|---|---|---|---|
| T1 | 83 - 53 - 70 - 45 - 44 | 82 - 8 - 44 - 19 - 2 | 76 - 48 - 62 - 80 - 29 | 76 - 82 - 43 - 16 - 84 | 72 - 33 - 58 - 2 - 55 |
| T2 | 39 - 22 - 80 - 10 - 0 | 73 - 37 - 89 - 67 - 18 | 99 - 60 - 89 - 39 - 69 | 95 - 78 - 91 - 30 - 22 | 84 - 54 - 75 - 28 - 40 |
| T3 | 18 - 30 - 73 - 33 - 90 | 4 - 92 - 83 - 24 - 14 | 14 - 74 - 59 - 87 - 55 | 1 - 96 - 25 - 81 - 62 | 39 - 15 - 41 - 12 - 35 |
| T4 | 4 - 76 - 77 - 12 - 31 | 93 - 90 - 84 - 81 - 66 | 40 - 46 - 54 - 92 - 7 | 5 - 18 - 63 - 14 - 24 | 23 - 49 - 91 - 32 - 38 |
| T5 | 55 - 88 - 26 - 42 - 69 | 40 - 72 - 56 - 36 - 51 | 6 - 32 - 77 - 27 - 63 | 23 - 75 - 9 - 60 - 27 | 64 - 68 - 6 - 92 - 18 |
| T6 | 15 - 40 - 96 - 9 - 72 | 50 - 68 - 88 - 55 - 57 | 96 - 33 - 49 - 25 - 68 | 83 - 20 - 90 - 55 - 36 | 48 - 47 - 13 - 89 - 79 |
| T7 | 11 - 47 - 85 - 28 - 93 | 27 - 29 - 80 - 3 - 94 | 26 - 94 - 38 - 85 - 98 | 4 - 10 - 77 - 93 - 33 | 96 - 22 - 34 - 81 - 63 |
| T8 | 5 - 66 - 65 - 35 - 16 | 53 - 62 - 87 - 52 - 95 | 61 - 43 - 93 - 15 - 28 | 58 - 35 - 97 - 11 - 59 | 53 - 85 - 14 - 50 - 44 |
| T9 | 49 - 34 - 7 - 95 - 27 | 70 - 12 - 1 - 97 - 48 | 36 - 2 - 42 - 75 - 31 | 56 - 98 - 47 - 86 - 38 | 24 - 61 - 11 - 0 - 21 |
| T10 | 19 - 81 - 25 - 62 - 13 | 60 - 47 - 65 - 10 - 41 | 22 - 56 - 3 - 67 - 19 | 85 - 66 - 49 - 41 - 87 | 10 - 59 - 90 - 71 - 56 |
| T11 | 24 - 3 - 17 - 38 - 8 | 17 - 96 - 9 - 49 - 30 | 20 - 90 - 50 - 84 - 66 | 42 - 99 - 57 - 0 - 6 | 17 - 76 - 1 - 95 - 70 |
| T12 | 78 - 6 - 64 - 36 - 89 | 38 - 58 - 0 - 26 - 21 | 70 - 97 - 4 - 64 - 44 | 70 - 13 - 50 - 40 - 68 | 94 - 37 - 5 - 4 - 26 |
| T13 | 56 - 99 - 54 - 43 - 50 | 31 - 15 - 75 - 25 - 6 | 82 - 47 - 95 - 41 - 51 | 48 - 73 - 37 - 8 - 39 | 60 - 20 - 45 - 98 - 74 |
| T14 | 67 - 46 - 68 - 61 - 97 | 74 - 59 - 64 - 43 - 34 | 23 - 5 - 79 - 88 - 34 | 32 - 3 - 89 - 51 - 44 | 62 - 57 - 73 - 97 - 87 |
| T15 | 79 - 41 - 58 - 48 - 98 | 20 - 77 - 7 - 78 - 71 | 16 - 35 - 52 - 71 - 72 | 17 - 54 - 15 - 67 - 2 | 46 - 51 - 7 - 82 - 83 |
| T16 | 57 - 75 - 32 - 94 - 59 | 22 - 39 - 63 - 76 - 85 | 57 - 12 - 1 - 13 - 86 | 31 - 52 - 61 - 34 - 71 | 19 - 88 - 9 - 8 - 52 |
| T17 | 63 - 84 - 37 - 29 - 1 | 79 - 45 - 61 - 42 - 46 | 78 - 8 - 21 - 91 - 83 | 64 - 92 - 65 - 53 - 28 | 30 - 65 - 16 - 36 - 69 |
| T18 | 52 - 21 - 2 - 23 - 87 | 54 - 91 - 16 - 5 - 33 | 10 - 0 - 65 - 73 - 37 | 72 - 80 - 12 - 45 - 21 | 25 - 67 - 43 - 29 - 42 |
| T19 | 91 - 74 - 86 - 82 - 20 | 35 - 98 - 69 - 32 - 99 | 45 - 30 - 17 - 53 - 58 | 29 - 7 - 26 - 79 - 69 | 78 - 80 - 31 - 86 - 93 |
| Downstream task | 60 - 71 - 14 - 92 - 51 | 86 - 23 - 13 - 11 - 28 | 11 - 9 - 81 - 24 - 18 | 94 - 74 - 46 - 19 - 88 | 77 - 27 - 99 - 66 - 3 |

Table 5: Task splits used in the results with Split MiniImagenet. The indices correspond to the classes listed in Table 4. Results reported on Split MiniImagenet average over these 5, randomly determined, task sequences.

