# OpenReview forum: "Knowledge Accumulation in Continually Learned Representations and the Issue of Feature Forgetting"
_TMLR — Accepted by TMLR_

### Review · Reviewer_DdoQ · 2024-03-18

**Summary Of Contributions:**

The paper presents an empirical investigation of forgetting in task-based continual learning (CL). It highlights and distinguishes between forgetting in representation level and output level. The paper shows that forgetting occurs at representational level and affects generalization performance to future tasks. The paper introduces several new metrics for measuring forgetting to support its analysis. Lastly, it demonstrated that functional regularization appears to more effectively mitigate forgetting.

**Audience:**

Yes

**Broader Impact Concerns:**

N.A

**Claims And Evidence:**

No

**Requested Changes:**

The current paper requires significant revision:
1. Consider the more general setting of class-incremental CL.
2. Propose concrete solutions to tackle the forgetting issues identified in the paper.
3. Revise the relative forgetting metric for output level (see discussion above in Strength and Weakness). In my view, linear probing is sufficient to demonstrate representational forgetting. Its comparison with forgetting at output level is unnecessary and unclear.

**Strengths And Weaknesses:**

Strength

1. The paper presents a timely and nuanced discussion on CL forgetting, which offers potential insights to the community for developing new CL algorithms.

Weakness
1. The setting considered in this work is limited to task-based CL, a restrictive setting requiring task ID to be present at test time. Class-incremental CL is a more general setting and the focus of more recent works (e.g., [1, 2]).
2. The paper does not offer concrete proposals for mitigating forgetting based on the findings. While I agree with the general finding on forgetting at representational level, it is unclear how this issue can be tackled effectively. The paper in this sense appears incomplete and offers limited contribution.
3. Technically, $r_{j-1, j}$  for the relative forgetting metric appears problematic. For $r_{i, j}=OUT_{i, j}$, $i$ should be be equal or larger than $j$, since the classification head $\theta_{h_j}$ does not exist at $f_{j-1}$, which has not seen the task j. Thus it is unclear what the appropriate interpretation for OUT_{j-1, j} is. In contrast, $r_{j-1, j} makes sense only for linear probing since the classifier head is computed on-the-fly for each task.


[1] https://arxiv.org/abs/2210.05561

[2] https://arxiv.org/abs/2112.08654

---

> ### Author Response · Authors · 2024-04-05
>
> We thank the reviewer for regarding our work as timely and nuanced, and for sharing our perspective that the insights we provide aid the community in further development of continual learning. Changes we made with regard to your comments are shown in orange in the revised version of the paper.
>
> We now would like to address the weaknesses pointed out by the reviewer.
>
> \
> **The setting considered in this work is limited to task-based CL, a restrictive setting requiring task ID to be present at test time. Class-incremental CL is a more general setting and the focus of more recent works (e.g., [1, 2]).**
>
> We would like to kindly object to the notion that the task-based setting is limiting our study. We clarify this here and add a clarification in the paper in Section 2, paragraph 3.
> *During training*, the difference between “task-incremental” and “class-incremental” setting with a cross-entropy loss is whether or not previously observed classes are in the denominator of the loss (or, in other words, whether or not the output units of past classes are set to “active” when training on a new task). Task-incremental learning makes use of separate heads, so by default excludes previous classes from the loss. In class-incremental learning, there is a single head, such that by default the output for previous classes are included. However, recently it has become a standard to mask classes in the loss that clear are not present in the training batch, to avoid interference. This is known as the “label trick” [1]. Using this approach makes class-incremental training the same as task-incremental training. This approach is - for example - also used in the work “Learning to Prompt” [2] (see [here](https://github.com/google-research/l2p/blob/main/train_continual.py), line 271) that the reviewer suggested. With this trick, our training setup is the same as class-incremental training.
> *During evaluation*, we make use of the task ID to exclude confounding factors from our analysis. In the class-incremental evaluation, the evaluation task itself becomes harder with every new task. This is because the decision space becomes more complex as more classes need to be separated from each other. For our analysis, this would introduce a confound, because it would mean that we could no longer distinguish whether a loss of performance is due to feature forgetting or due to the increased complexity of the evaluation task.
>
> [1] “Task agnostic continual learning using online variational bayes” Zeno et al. - 2018
>
> [2] “Learning to Prompt” Wang et al. - 2022
>
> \
> **The paper does not offer concrete proposals for mitigating forgetting based on the findings. [...]**
>
> It is true that we do not propose concrete solutions. We believe it is important to clearly describe and analyze the problem, which has required plenty of experiments and critical thinking. We hope that this analysis can inspire other researchers to find solutions to prevent forgetting. In Section 6 we provide some ideas for future research, which we hope to see pursued in the future.
>
> \
> **Technically, $r_{j-1, j}$ for the relative forgetting metric appears problematic. [...]**
>
> We define the head of task $T_j$ at timestep $i < j$ (i.e., when it is not yet trained) as a randomly initialized head. We now clarify this in Section 2 (second paragraph) of the paper. Such a head results in random guessing performance (e.g. ~20% for a 5-way classification task), similar to random guessing. We think this is an appropriate baseline, also for output accuracy. Before training a task, a model does not have zero percent accuracy, it has random guessing performance. So any result below this would imply losing more performance than was initially gained. We can see in e.g. Figure 2 that output accuracy falls back to random guessing performance, not to zero percent. Additionally, using random guessing performance as a baseline reduces the influence of the number of classes in a task, as this is inversely proportional to random guessing performance.

---

### Review · Reviewer_xXFR · 2024-03-20

**Summary Of Contributions:**

The paper studies the question of representation forgetting in continual learning. The authors reevaluate the methodology used in previous papers of how to use linear probes to measure forgetting. In particular, they notice that the baseline linear probing performance is high even before starting training on a given task. As such, the absolute drop of linear probing accuracy is not high, even though the relative drop is quite significant. The authors then compare the performance of fine-tuning with an ensemble-like approach to demonstrate that there is in fact a significant gap in terms of knowledge accumulation. Finally, the authors check how different CL methods impact the dynamics of knowledge accumulation and forgetting.

**Audience:**

Yes

**Claims And Evidence:**

Yes

**Requested Changes:**

I suggested some additional experiments in the weaknesses section above, but it's more of a "nice-to-have". I think it is a solid study as it is.

**Strengths And Weaknesses:**

In my opinion, this is an excellent, carefully performed study of representation forgetting

Strengths:
- The paper studies an important problem, namely the forgetting of representations in CL.
- The main observations made in this paper are quite interesting and could be useful for further studies on continual learning.
- The experiments are thorough and, as far as I can tell, carefully done. I particularly appreciate the ablation of the ensemble experiment provided in Appendix B and additional experiments on Cifar-100 in Appendix F.
- To the best of my knowledge, the related work is properly discussed.
- The paper is clearly written and the authors properly explain their chain of thought.

Weaknesses:
- This is more of a “nice-to-have” than an actual weakness, but I would like to see more experiments on the experience replay in Section 5:
    - ER with 20 samples per class is not that many. I would be curious to see what happens as we increase the number of samples -- in the limit, we should achieve something similar to multi-task learning.
    - I think using ER + some kind of distillation could be also interesting (e.g. Dark Experience Replay) to see whether it impacts the representation forgetting. LwF already is a distillation method, but I think a more thorough investigation here could lead to further insights.

---

> ### Author Response · Authors · 2024-04-05
>
> We would like to thank the reviewer for their general appraisal of our work as an excellent, carefully performed study of representation forgetting. We appreciate the recognition of the importance of the problem we address, the thoroughness of our experiments, and the comments on the writing and composition of our work. Changes we made with regard to your comments are shown in pink in the revised version of the paper.
>
> \
>  **More replay experiments**
>
> We followed the suggestion of the reviewer and conducted additional experiments with increased buffer size for the replay and added DER (Dark Experience Replay) to test the combination of replay and distillation. The results can be found in the Appendix section E and a small discussion in Section 5.

---

### Review · Reviewer_VqjT · 2024-03-23

**Summary Of Contributions:**

The main contribution of this paper is to assess that when looking at performance increase and decrease when learning a task and then forgetting is linear probed performance forgetting is similar to output forgetting.
They propose to use the "relative forgetting" metric to assess their claim.
And they show that this is indeed the case for four approaches published before 2019.

This contribution put a grain of salt into recent publications that could lead to the idea that feature forgetting was less problematic than it could appear.

**Audience:**

Yes

**Claims And Evidence:**

Yes

**Requested Changes:**

While I believe this paper could be an interesting reading for anyone willing to understand the relationship between feature forgetting and catastrophic forgetting, the current state of the paper is misleading toward the positioning of the paper in the existing literature.

Hence, the positioning of the paper should be reworked because it currently does not reflect where the paper stands in the space of continual learning literature.
First, the literature opinion should not be assumed from a couple of papers.
While I think it is important to have new articles adding a grain of salt to recent publications, the link between such articles and other literature has to be made more in-depth and more clearly.
In my understanding, the paper aims at clarifying and quantifying the distinction between feature forgetting and catastrophic forgetting, which is important.
However, it does not discover that feature forgetting is a thing that prevents knowledge accumulation, because that is what literature on task incremental learning is about.
Consequently, my change request is essentially to make more clear that this paper is about understanding better claims about low feature forgetting and existing literature.

Also, a slight rework on introducing/concluding each part would be necessary to follow easily how each section relates to each other.

**Strengths And Weaknesses:**

**Strengths:**

The paper is well-written and quite easy to follow.

I like the figure 2, which makes the point of this "relative forgetting" metric quite clear. I think it is interesting to ponder forgetting by the relative performance gain that the task provides, which might be not significant for LP.

The figures are well explainable of what the authors aim to show.

The experiments are well conducted and are convincing to make the point of the authors.

**Weaknesses:**

general weaknesses:
- the related work offers a list of related literature however it is not used as context for the reader to understand what the proposed paper is about. There is no link made between the literature and the proposed contribution, which misses the point of related work. Also as a paper about knowledge accumulation (as the title suggests currently) related work section on knowledge accumulation would make sense.
For example, the paper cites,  [Caccia et al., 2022] paper which is about representation drift, it sounds like it is a closely related paper that the reader should be aware of as context for reading the whole paper.


- it is not always clear how each section serves the purpose of the paper, I missed it for section 4.2 for example. Some small introductions/conclusions for subsections would help to keep the reader on board with how each section is articulated to serve a specific purpose.

Writing weaknesses:
Some formulations are a bit vague and should be made clearer in particular when it is in the abstract.

- [abstract] > Multiple recent studies ascribe representations a certain level of innate robustness against forgetting - that they only forget minimally and no critical information.
-> The whole sentence and in particular "no critical information" sounds like an overclaim about what is written in the literature here, I believe only one of the cited paper really claim such a bold thing while the others just claim that forgetting is quite limited with respect to usual catastrophic forgetting.

- [abstract] >  "Even though it is true that feature forget-
ting can be small in absolute terms, newly learned information tends to be forgotten just
as catastrophically at the level of the representation as it is at the output level"
-> unclear, does the author want to make a distinction between initially learned features and the one that are learned later? What is the relationship between information, features and representation here?

-> One claimed contribution is "We show that forgetting of the representation hinders knowledge accumulation". I do not understand, in such a formulation, how it could be otherwise. How forgetting could not be a knowledge accumulation issue? Can you comment on that, please?

- Also, in section 3, you are citing [Zang et al 2022] to say "had observed this upward trend and concluded that feature forgetting does not pose an issue for learning representations continually".
-> One of the listed contributions of [Zang et al 2022] is "We show interpretable dynamics of feature learning and that feature forgetting can significantly impair performance". Can you comment on that? In my understanding, [Zang et al 2022] claims that "representation learning and catastrophic forgetting are largely separate issues", they do not say representation forgetting is not an issue.

- [4.3] "A better representation can make learning easier"
-> I think a better representation that makes learning harder would not be better anymore, what is the point of this claim?


- [Discussion] "It is possible to think of two tasks that have nothing in common and thus have no knowledge accumulation"
->Learning a new task that brings no improvement to the former is not proof that the model does not have more knowledge than before.
If your concept of knowledge accumulation assumes that indeed such a case is a situation of 'no knowledge accumulation' this concept should be renamed because it is misleading.


- [4.3] "Previous sections showed that there is severe feature forgetting"
-> I think saying that "feature forgetting severely impacts newly learned features" would be more in line with what was shown in the previous section.




Positioning weaknesses:

My biggest concern is about the positioning of the paper in the literature.
Beyond the fact that the related work is in the end and is not used to put the paper in the context of the literature.

- Despite the few papers cited in the introduction by the authors (Davari et al., 2022; Zhang et al., 2022; Hu et al., 2022; Wang et al., 2023),
the whole literature on task incremental learning shows the issue of feature forgetting.
Task incremental learning is only about feature forgetting.
How does the author experiment in section 5 show anything different than what task incremental learning would show?


- [related work] The paper title starts with "Knowledge Accumulation in Continually Learned Representation" A section on knowledge accumulation in related work should appear. At least, to compare definitions of knowledge accumulation and the difference of approaches, for example in "Challenging Common Assumptions about Catastrophic Forgetting and Knowledge Accumulation" Lesort et al, CoLLas 2023, or in [Caccia et al 2020] that you cite in section 7.

---

> ### Author Response · Authors · 2024-04-05
>
> We would like to thank you for referring to our paper as an interesting read to understand the relation between feature forgetting and catastrophic forgetting, with well conducted and convincing experiments. Your extensive review and comments helped us improve the overall clarity and structure of the paper. The changes we made with regard to your comments are shown in blue in the revised version of the paper.
>
> \
> **The related work offers a list of related literature however it is not used as context for the reader [...]**
>
> We added more related work to Section 3, especially on knowledge accumulation, see also below. We kept the other related work at the end, because we think they are interesting background information - for readers that want to delve deeper in the subject - but not necessary to follow our paper. We renamed this section to ‘Background’ and added a clarification to its introduction.
>
> **Also as a paper about knowledge accumulation (as the title suggests currently) related work section on knowledge accumulation would make sense. For example, the paper cites, [Caccia et al., 2022] paper which is about representation drift [...]**
>
> We included more related works on knowledge accumulation and a discussion on representation drift  in Section 3. It is indeed important; it clearly shows that there are more sources of forgetting than feature forgetting when looking at the output accuracy.
>
> **it is not always clear how each section serves the purpose of the paper, I missed it for section 4.2 for example.[...]**
>
> We added some more clarifications to the beginning of section 4.2, as well as a general introduction and overview of Section 4. We hope this helps the overall clarity of the paper.
>
> **[abstract] > [...] The whole sentence and in particular "no critical information" sounds like an overclaim [...]**
>
> This was indeed a bit strong. We updated this sentence in the abstract. See also “on the connection to literature” below.
>
> **[abstract] > "Even though it is true [...]" -> unclear, does the author want to make a distinction between initially learned features and the one that are learned later? What is the relationship between information, features and representation here?**
>
> See updated sentence in the abstract. Information is indeed ambiguous, as besides information that improves the current task, there might be other information too. The representation consists of features, we further clarified that in the abstract.
>
> **[...] How forgetting could not be a knowledge accumulation issue?[...]**
>
> See “relation of KA to feature forgetting” below.
>
> **Also, in section 3, you are citing [Zang et al 2022] [...], [Zang et al 2022] claims that "representation learning and catastrophic forgetting are largely separate issues", they do not say representation forgetting is not an issue.**
>
> See “relation of KA to feature forgetting” below.
>
>
> **[4.3] "A better representation can make learning easier" [...]**
>
> This sentence is indeed ambiguous. We updated the text to better reflect what we intended to say; that a representation that is already good at solving a new, unseen, task needs to change less than a representation that performed worse. If less changes are necessary, this likely reduces the risk of forgetting.
>
> **[Discussion] [...] Learning a new task that brings no improvement to the former is not proof that the model does not have more knowledge than before.[...]**
>
> Using knowledge accumulation here is indeed not exactly what we mean. We updated this sentence. See also the new information on knowledge accumulation below and in Section 3.
>
> **[4.3] "Previous sections showed that there is severe feature forgetting" [...]**
>
> Agreed, this is more accurate. We changed this sentence.
>
> **Despite the few papers [...] How does the author experiment in section 5 show anything different than what task incremental learning would show?**
>
> See “On the relation to task incremental learning” and “in relation to Section 5” below.
>
> **[related work] [...] A section on knowledge accumulation in related work should appear. [...].**
>
> See new information on Knowledge Accumulation below and in Section 3.

---

> > ### Author Response · Authors · 2024-04-05
> >
> > **On the relation with task incremental learning.** There is a connection between our work and task incremental learning. In Figure 2, the results of the output accuracy are those of a task incremental experiment. We agree that task incremental learning is in part about knowledge accumulation, because if it was not, one could also train an individual model for each task. We think it is still important to show that feature forgetting hinders knowledge accumulation the way we do. In task incremental learning, forgetting has two parts. First, there is forgetting in the representation (i.e. the representation of the task data changes such that the classes of this task become less separable than before, even with the best possible head) and secondly, forgetting in the head itself (i.e. the representation of the data is separable, but the head does not represent the best possible decision boundaries). In this paper, our goal is to give more insight into the first source of forgetting, i.e. feature forgetting. In task incremental learning, both sources of forgetting are studied together. Results in task incremental learning do not guarantee that there is no knowledge accumulation in the representation, it is possible that the head does not find the correct decision boundaries, even though the representation got better. Our experiment unambiguously shows that feature forgetting prevents knowledge accumulation in the representation, and that not only the head is the problem. We added this explanation to Section 3 on feature forgetting.
> >
> > **The connection to literature.** Our goal in the introduction was to quote the papers that, like ours, study forgetting of the representation directly. As far as we are aware, these are rather complete. We will include more if the reviewer is aware of other papers that study this problem. We did not discover feature forgetting, if we made such claims in the paper, we will remove those. We study the “characteristics of feature forgetting” (contribution 1) and show that “feature forgetting is equally bad at the level of representation as at the output when considering newly gained performance” (contribution 2), which is in contrast to what the earlier works claim, but we did not discover feature forgetting.
> >
> > **In relation to Section 5.** The main idea of this section is to show the effect of different methods on representation forgetting. The results show that for most methods, (relative) forgetting in the representation is at the same level as at the output. Yet there is more; e.g. for replay, we can see that there is more forgetting in the representation than the head. This suggests that standard replay in a large part helps by finding a good head for a particular representation, but not necessarily for the representation.
> >
> > **Knowledge Accumulation.** With Knowledge Accumulation (KA), we refer to improved performance on a task that was not currently trained (i.e. a downstream task), which measures the generalization capabilities to new and unseen tasks, which is how this term is used in e.g. [Jin et al. 2021.]. Related but not exactly similar, KA is used in [Lesort et al. 2023; Caccia et al. 2020] in a meta-learning context, where representations that can easily adapt to new (or to older and forgotten) tasks are the goal. These meta-tasks are similar to our downstream tasks. The goal of continual pre–training is also related; learning representations that generalize well to unseen tasks, but they are not concerned with feature forgetting of the pretrained data [Hu et al. 2022; Lee et al 2023; Cossu et al. 2022]. Lastly, as explained above, task incremental learning is at least in part about knowledge accumulation as well, see e.g. [De Lange et al. 2022] who list forward transfer as one of the top desiderata for this setting. Yet task incremental results themselves are not well suited to study the representations directly, as representation drift [Caccia et al. 2021] can result in misaligned heads, which could suggest that there is no knowledge accumulation, even when there is.
> >
> > **Relation of KA to feature forgetting.** Forgetting features that were learned to solve a particular task will always impact the results of that particular task, but not necessarily those of downstream tasks. The learned and forgotten features might possibly be irrelevant for (most) downstream tasks. E.g. a feature encoding the shape of a tree leaf is probably only useful to categorize trees, not for other tasks. With knowledge accumulation we refer to improved generalization to unseen tasks, for which forgetting of a very task specific feature (like the shape of a tree leaf) might not impact the results. In the experiments of Section 4.3, we show that this is not generally true, and not forgetting any features leads to the best representation, which is in contrast to [Zang et al 2022], which we also interpret to claim that "representation learning and catastrophic forgetting are largely separate issues".

---

### Decision · Action_Editor_BsKJ · 2024-05-01

**Recommendation:** Accept with minor revision

**Comment:**

Two of the three reviewers originally recommended that the paper be accepted, recognizing the quality of this study. We discussed this further with the third reviewer, who seems to agree that the work meets the TMLR criteria.

Overall, the main points of discussion with (two of) the reviewers were whether the findings were previously known to the community, given that current works in CL do not (explicitly) distinguish between "representation" vs. "output" forgetting. In other words, how important is this distinction? Of course, as the authors point out, recent works discard the importance of representation forgetting, which is partly what this paper clarifies. There was also a suggestion by reviewers to explain how the findings add to the literature on class-incremental learning (with functional regularization), which seems to deal with representation forgetting more directly.

I recommend this paper be accepted. I propose minor corrections since the authors might want to address the above issue further. I don't make it a requirement since these points have already been discussed in the paper and its revision. However, the fact that two (expert) reviewers still had comments after the discussion might be indicative that the positioning of the contribution with respect to the current literature could be improved. One suggestion would be to consider further clarifying/emphasizing using additional references, but again, this is at the authors' discretion.

Congratulations!

**Audience:**

All reviewers agree that it would interest the TMLR audience, particularly those interested in continual and transfer learning.

**Claims And Evidence:**

After the internal discussion, the conclusion was that the claims were correctly supported by evidence. I give more details about it below.